# Learning Evolving Tools for Large Language Models

**Guoxin Chen**[1], **Zhong Zhang**[2*], **Xin Cong**[2*],
**Fangda Guo**[1], **Yesai Wu**[2], **Yankai Lin**[3], **Wenzheng Feng**[4], **Yasheng Wang**[4]
[1]Institute of Computing Technology, Chinese Academy of Sciences
[2]Tsinghua University [3]Renmin University of China [4]Huawei Noah's Ark Lab
chenguoxin22@mails.ucas.ac.cn
{zhongzhang,congxin1995}@tsinghua.edu.cn

## ABSTRACT

Tool learning enables large language models (LLMs) to interact with external tools and APIs, greatly expanding the application scope of LLMs. However, due to the dynamic nature of external environments, these tools and APIs may become outdated over time, preventing LLMs from correctly invoking tools. Existing research primarily focuses on static environments and overlooks this issue, limiting the adaptability of LLMs in real-world applications. In this paper, we propose TOOLEVO, a novel framework designed to enhance the adaptive and reflective capabilities of LLMs against tool variability. By leveraging Monte Carlo Tree Search, TOOLEVO facilitates active exploration and interaction of LLMs within dynamic environments, allowing for autonomous self-reflection and self-updating of tool usage based on environmental feedback. Additionally, we introduce ToolQA-D, a benchmark specifically designed to evaluate the impact of tool variability. Extensive experiments demonstrate the effectiveness and stability of our approach, highlighting the importance of adaptability to tool variability for effective tool learning.[1]

## 1 INTRODUCTION

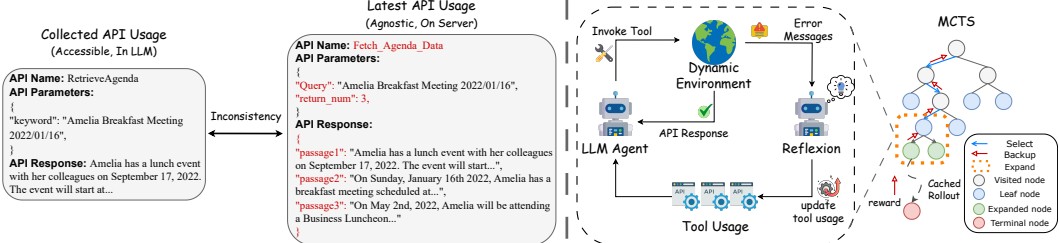

Figure 1: (Left) An example of inconsistent usage (name, parameters, or response formats) between the collected APIs available to LLMs and the latest APIs deployed on the server. The collected APIs may become outdated over time. (Right) An overview of our TOOLEVO. The LLM engages with the dynamic environment using MCTS for fine-tuning against tool variability, reflecting and updating tool usage based on environmental feedback. Each node in MCTS contains an API invocation.

Tool learning aims to augment large language models (LLMs) with tools and APIs[2] (Schick et al., 2023; Qin et al., 2023a; Guo et al., 2024). This augmentation enables LLMs to interact with external environments and real-world applications, thereby expanding their capabilities to tackle a diverse array of complex tasks (Qiao et al., 2024b; Yang et al., 2024b; Lu et al., 2024) and having become a vital component in building LLM agents (XAgent, 2023; Chen et al., 2024c; Qian et al., 2024).

A critical challenge often overlooked is the inherent dynamism of external environment. In the realm of tool learning, dynamic environments are primarily manifested as **tool variability**, which include changes in API names, parameters, or response formats. APIs are subject to continual evolution due to various factors, such as version updates, optimizations, and deprecations. This rapid and frequent

---

*Corresponding author.
[1]Our code is available at https://github.com/Chen-GX/ToolEVO.
[2]We use the term tools and APIs interchangeably.

change of tools is difficult to capture in a timely manner. As a result, there is a discrepancy between the APIs that LLMs have learned to use and those deployed in the real-world environment, which leads to LLMs being unable to correctly invoke the tools, as illustrated in Figure 1 (left).

Typical tool learning approaches first fine-tune the LLMs with massive tool usage data to learn the behavior of tool invocation. Then, during the inference stage, they provide the required tools for the tasks by offering a tool manual through zero-shot prompting or demonstrating tool usage through few-shot prompting (Qin et al., 2023b; Wang et al., 2024; Yang et al., 2024b). However, this paradigm presents a serious risk: if the specified tools in the prompt do not keep pace with changes in the external environment, the model will incorrectly invoke the outdated APIs instead of the latest APIs, leading to a collapse in performance, as shown in Figure 2. A straightforward solution to this problem is to collect and update the latest APIs in real time, which is time-consuming and resource-intensive. Therefore, in this work, we aim to improve the model's adaptability in tool learning to better handle the complexities of dynamic external environments.

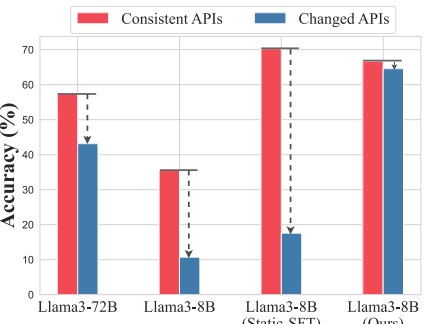

To address the above issues, we propose TOOLEVO, a novel framework that facilitates the adaptive and reflective capabilities of LLMs through active exploration against tool variability. Inspired by mature biological systems such as humans (Johnson-Frey, 2003; Smitsman et al., 2005), we posit that the key to adapting to tool variability lies in active interaction with the dynamic environment, coupled with self-reflection and self-updating of tool usage from trial and error. As shown in Figure 1 (right), we actively expose LLMs to dynamic environments, enabling autonomous adaptation to tool variability. Specifically, TOOLEVO manages the extensive action space in dynamic environments using Monte Carlo Tree Search (MCTS), while reflecting on and updating existing tools usage—which may be outdated—based on environment feedback. This approach allows LLMs to understand tool variability through fine-tuning on these autonomously explored trial and error, rather than merely memorizing the invocation patterns of existing tools. Furthermore, for research purposes, we construct a new

Figure 2: Impact of tool variability. "Consistent APIs" refer to APIs that are consistent between LLMs and servers. "Changed APIs" refer to APIs accessible to LLMs that are outdated over time. "Static-SFT" is supervised fine-tuning on tool usage data that has no adaptability to tool variability. Our method successfully adapts to API changes.

benchmark ToolQA-D based on ToolQA (Zhuang et al., 2023b) to investigate the impact of tool variability. Extensive experiments have significantly demonstrated the effectiveness and stability of our approach in adapting to tool variability. Our contributions are summarized as follows:

- To our knowledge, we are the first to investigate the impact of tool variability on the performance of LLMs, which is crucial for ensuring their reliability and adaptability in real-world applications.
- We propose a self-adaptive framework TOOLEVO, designed specifically to address tool variability. Specifically, we actively enhance the interaction and exploration of LLMs within dynamic environments through MCTS. In this way, TOOLEVO empowers LLMs to understand tool variability through trial and error, rather than merely replicating existing tool invocation patterns.
- For research purposes, we have constructed the first benchmark for tool variability, ToolQA-D. Based on this benchmark, we comprehensively analyze the impact of various API changes (such as names, parameters, and response formats) on LLMs, thereby promoting further research.
- Through extensive experiments under various tool variability settings, we demonstrate the effectiveness of our approach and highlight the importance of adaptability for effective tool learning.

## 2 PRELIMINARIES

### 2.1 TASK FORMULATION

In this paper, we consider the scenario of tool variability, which is both prevalent and demanding in practical applications. The input for this task consists of the task description $\mathcal{D}$ and the collected APIs $\mathcal{P}_c = \{\mathcal{P}_c^1, \mathcal{P}_c^2, \ldots, \mathcal{P}_c^n\}$. However, due to tool variability, the APIs actually deployed on the server, denoted as $\mathcal{P}_s = \{\mathcal{P}_s^1, \mathcal{P}_s^2, \ldots, \mathcal{P}_s^m\}$, may differ from $\mathcal{P}_c$ in terms of API names, parameters

or response formats. In practice, the environment will provide various types of feedback, defined as $\mathcal{O}$, including task completion states, API responses, and API error messages. The objective of this task is to successfully complete tasks under tool variability based on environmental feedback.

## 2.2 DEFINITIONS OF KEY ELEMENTS IN MCTS

MCTS (Shapiro, 2003; Browne et al., 2012) is a heuristic search algorithm designed for decision processes, particularly suited for scenarios with large action spaces and uncertain outcomes in dynamic environments. In the context of our proposed TOOLEVO, MCTS plays a pivotal role in encouraging LLMs to actively engage with dynamic environments. We formulate the tool variability task within MCTS as a multi-step Markov decision process (MDP) (Puterman, 1990) in which:

- **State** $s_t \in \mathcal{S}$: represents the current context of the LLM's operational environment, which consists of the task description $\mathcal{D}$, the currently available API usage $\mathcal{P}_c$, all actions and its environment feedback taken along the search path from the root node to the current node. The MCTS begins with an initial state $s_0$, which includes the input task $\mathcal{D}$ and the collected API usage $\mathcal{P}_c$.
- **Action** $a_t \in \mathcal{A}$: defined as an API invocation in the REACT (Yao et al., 2023) format, which consists of thought (textual analysis), tool invocation (APIs), and observation (environmental feedback). A detailed template is listed in Appendix A.1. We employ the LLM to generate actions at each state, represented as $\pi_\theta(a_t|s_t) = \texttt{LLM}(a_t|s_t)$, leading to a transition to a new state $s_{t+1}$ by concatenating $s_t$ and $a_t$, such that $s_{t+1} = \texttt{Cat}(s_t, a_t)$.
- **Dynamic Environment**: In alignment with real-world scenarios, we define the dynamic environment as everything excluding the LLM agent. Firstly, the APIs $\mathcal{P}_s$ deployed in the environment may differ from the APIs $\mathcal{P}_c$ provided to the LLM. Secondly, the dynamic environment will provide the following information (Robbes et al., 2012; Brito et al., 2018):
  - **Task Completion State**: Upon task completion, the environment provides feedback indicating whether the task was successful (reward $r = 1$) or failed (reward $r = -1$). Specifically, we assess task completion state through the evaluation toolkit (Zhuang et al., 2023b). We assign rewards to the search path (tool trajectories) based on the task completion state.

$$r = \begin{cases} 1 & \text{if task is successful} \\ -1 & \text{if task is failed} \end{cases} \tag{1}$$

  - **API Response**: The API response refers specifically to the information returned following a successful API invocation. Different APIs provide various functionalities, thereby enabling LLMs to perform complex tasks.
  - **API Error Message**: API error message can be categorized into two main types: invocation errors and deprecation errors.
    * **Invocation Errors**: These errors arise from incorrect API names or parameter settings and are not related to API deprecation. To address such errors, the model must possess self-reflective capabilities, allowing it to adjust its input to successfully invoke the API.
    * *Deprecation Errors*: These indicate that an API has been removed in the current version, with a recommendation to utilize a newer API instead. To address such errors, the model must process tool update capabilities based on environment feedback.

## 3 METHODOLOGY

### 3.1 OVERVIEW

As illustrated in Figure 1 (right), we propose the TOOLEVO framework, which aims to actively immerse the LLM in dynamic environments with the help of MCTS. This enables the LLM to autonomously reflect on and update existing tool usage, rather than merely executing rigid tool invocations. Through actively exploring dynamic environments, the LLM accumulates trial-and-error experiences for fine-tuning, enhancing its understanding and adaptability to tool variability.

### 3.2 INTERACTION WITH DYNAMIC ENVIRONMENTS

Considering the risk that the API usage $\mathcal{P}_c$ provided in the prompt may become outdated over time, our TOOLEVO encourages the LLM to interact with dynamic environments through MCTS,

thereby enhancing its adaptive and reflective capabilities regarding tool variability. Specifically, we customize the four key operations of MCTS as follows:

**Selection:** During the selection process, we traverse the tree from the root to a leaf node using the PUCT algorithm (Rosin, 2011). This algorithm selects the most promising node to explore, striking a balance between exploring new states and exploiting known valuable states, which can be represented as:

$$\text{PUCT}(s, a) = Q(s, a) + c_{\text{puct}} \cdot P(s, a) \frac{\sqrt{N(s)}}{1 + N(s, a)}, \tag{2}$$

where $c_{\text{puct}}$ is a constant that controls the balance between exploration and exploitation; $Q(s, a)$ and $P(s, a)$ are the $Q$-value and prior probability of taking action $a$ in state $s$, respectively. Additionally, $N(s)$ and $N(s, a)$ denote the visit counts for state $s$ and for taking action $a$ from state $s$, respectively.

**Expansion:** Once an expandable leaf node is selected, the current state $s_t$ is used as input to further expand the tree. Candidate actions are sampled using the policy $\pi_\theta$ (*i.e.*, the LLM), as follows:

$$a_t^1, a_t^2, \ldots, a_t^k = \pi_\theta(a|s_t), \tag{3}$$

where $k$ is the number of expansion nodes. Each action $a_t^i$ represents either an API invocation or an update in API usage after self-reflection, as detailed in Section 3.3. Upon executing the corresponding API, the expanded new state $s_{t+1}^i = \text{Cat}(s_t, a_t^i)$ is derived from the current state $s_t$.

**Simulation (Cached Rollout):** In the simulation step, a rollout strategy is employed to simulate a complete episode from one of the newly added nodes $s_{t+1}^i$ to obtain an accurate reward $r$ (Silver et al., 2016). Inspired by Bellman et al. (2015) and He et al. (2023), we propose a cached rollout strategy to enhance efficiency, as detailed in Appendix A.3. In our cached rollout strategy, each node (including state, action, reward, and so on) in the simulated episode is cached in the tree. However, these cached nodes remain invisible to the tree during the selection phases. Before each expansion or simulation step, the cache is checked for the current state to either reuse stored results if available or perform a new expansion or simulation as needed. This approach reduces redundant calculations and significantly improves efficiency by avoiding the need to rollout from scratch in every iteration.

**Backpropagation:** After the simulation, the reward $r$ is propagated back along the path from the selected leaf node to the root, updating the visit counts $N$ and the $Q$-values of these nodes as follows:

$$Q(s, a) \leftarrow Q(s, a) + \frac{1}{N(s, a) + 1}\big(r - Q(s, a)\big); \quad N(s, a) \leftarrow N(s, a) + 1. \tag{4}$$

By applying these updates, MCTS incrementally refines its search strategy based on feedback from the dynamic environment, leading to more accurate and informed decisions in subsequent selections.

### 3.3 SELF-REFLECTION AND TOOL-UPDATE

With the customized MCTS described above, our TOOLEVO actively engages LLMs in both exploration and exploitation within dynamic environments. To address tool variability, we incorporate the self-reflection and tool-update module into MCTS based on environmental feedback. This allows the LLM not only to deal with invocation errors through self-reflection but also to autonomously summarize new tool usage through the tool-update module to update API usage $\mathcal{P}_c$ in the prompt, ultimately fostering a deeper understanding and robustness in the face of tool variability.

**Self-Reflection**  In our TOOLEVO, the self-reflection module generates verbal reflections based on external environmental feedback, providing valuable insights for future corrections. It is important to note that, as shown in Figure 1, the self-reflection module and the LLM agent are the same LLM and we do not rely on a more powerful model, such as GPT-4. The external environment typically provides various API error messages, including invocation errors and deprecation errors (Robbes et al., 2012; Brito et al., 2018). The purpose of encouraging LLM to interact with the external environment through MCTS is to enable the model to effectively utilize the environmental feedback, rather than backtracking or stopping when encountering errors as in previous work (Qin et al., 2023b; Guo et al., 2024). Specifically, when an error message is encountered, we use this error state $s_t$ as input, allowing the model to reflect on the cause of the error and try to solve it.

$$a_{\text{ref}_t}^1, a_{\text{ref}_t}^2, \ldots, a_{\text{ref}_t}^k = \pi_\theta(a|s_t); \quad s_{t+1}^i = \text{Cat}(s_t, a_{\text{ref}_t}^i), \tag{5}$$

where $a_{\text{ref}_t}$ is the reflective action based on the error state $s_t$. We will append these new reflective states $s_{t+1}^i$ after $s_t$ instead of interrupting further exploration of this error state.

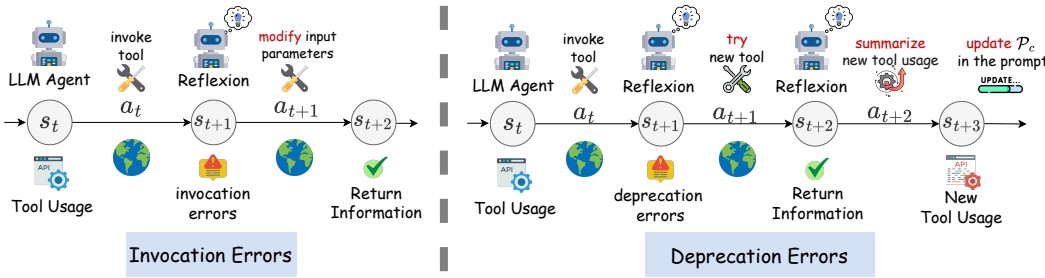

Figure 3: Examples of self-reflection and tool update. Invocation errors indicate that the input parameters of the API need to be corrected. In contrast, deprecation errors suggest that the input parameters are correct, but the API has been deprecated, necessitating an update in API usage.

**Tool-Update**   The self-reflection module analyzes the causes of error states based on various environmental feedback, such as API invocation errors. In the context of tool variability, we specifically focus on errors arising from API changes, referred to as deprecation errors (Zhou & Walker, 2016). These errors highlight inconsistencies between the API usage referenced in the prompt and the API usage deployed on the server. Unlike invocation errors, deprecation errors indicate that the invoked API is outdated and require updating the API usage (Sawant et al., 2018).

Therefore, we propose a tool update module based on self-reflection, as illustrated in Figure 3. Our tool update module requires the LLM to summarize the usage of new tools and incorporate them into $\mathcal{P}_c$ after successfully invoking the new tools based on environmental feedback. This process allows the LLM to gradually adapt to the dynamic tool environment ($\mathcal{P}_c \rightarrow \{\mathcal{P}_c + \mathcal{P}_s\}$) during exploration. Specifically, we implement the tool update module as a system tool, referred to as `UpdateTool[newtool_desc]`, which serves to update the descriptions of existing tools (Appendix A.5). In this context, the tool update module acts like a memory mechanism (Zhang et al., 2024), enabling the model to summarize and modify the API usage in the prompt based on newly acquired knowledge. This enhancement not only helps improve the accuracy of subsequent tool invocations but also fosters a more seamless integration of new tools into the LLM's workflow.

### 3.4   SELF-IMPROVEMENT FROM TRIAL AND ERROR

In our TOOLEVO, we employ MCTS to encourage the LLM to interact with dynamic environments, thereby collecting trial-and-error experiences (trajectories from the root node to the terminal node in the tree) involving self-reflection and tool updating. Through these experiences, we aim for the model to master the behavior of self-reflection and self-updating of tool usage in a dynamic environment, thereby enhancing adaptability to tool variability rather than merely learning the specific tool invocation. We evaluate the quality of these experiences based on whether the task is successfully completed, allowing the LLM $\pi_\theta$ to improve itself through the successful experiences $y^+$:

$$\mathcal{L} = \arg\min_\theta - \log \pi_\theta(y^+ | \mathcal{D}, \mathcal{P}_c). \tag{6}$$

Notably, failed experiences can be further leveraged through preference learning (Rafailov et al., 2023). However, our work emphasizes the importance of interaction with dynamic environments. We focus solely on SFT for a fair comparison following the prior research (Zhuang et al., 2023b).

## 4   TOOLQA-D

To our knowledge, we are the first to investigate the impact of tool variability on tool learning, leading to the absence of benchmarks in this domain. For research purposes, we have constructed the first benchmark for tool variability, termed ToolQA-D[3], based on ToolQA (Zhuang et al., 2023b). We define the original APIs of ToolQA (Zhuang et al., 2023b) as $\mathcal{P}_c$ (collected APIs), which may be outdated. For simulating tool variability, we employ GPT-4 to randomly modify the collected API usage, including names, parameters, and response formats. This results in two new sets of API usage: $\mathcal{P}_{s_{\text{in}}}$ and $\mathcal{P}_{s_{\text{OOD}}}$. The purposes of 3 different sets of API usage are outlined as follows:

---

[3]There are several compelling reasons in Appendix B for developing this dataset based on ToolQA

- On the prompt side:

  For all methods, $\mathcal{P}_c$ is used as to demonstrate tool usage.
- On the server side:
  - Deploying $\mathcal{P}_c$ represents the most common setup in existing studies. This setup maintains the static environment where prompt-provided APIs are consistent with server-deployed APIs.
  - Deploying $\mathcal{P}_{s_{in}}$ introduces variability in the tools. This setup allows us to collect trial-and-error experiences in a dynamic environment.
  - Deploying $\mathcal{P}_{s_{OOD}}$ also introduces tool variability but in an out-of-distribution context. The training data and prompts for all methods do not contain any API information from $\mathcal{P}_{s_{OOD}}$.

Ultimately, our ToolQA-D comprises 7 datasets and 3 sets of API usage ($\mathcal{P}_c$, $\mathcal{P}_{s_{in}}$ and $\mathcal{P}_{s_{OOD}}$), accompanied by a total of 6,234 and 5,884 training samples, 700 and 700 development samples, and 700 and 730 test samples for Easy and Hard difficulty respectively.

## 5 EXPERIMENTS

### 5.1 EXPERIMENTAL SETUP

**Dataset Setup** We conduct extensive experiments on ToolQA-D to investigate tool variability. In this work, we aim to demonstrate the importance of enabling LLMs to interact with dynamic environments during the training phase, rather than merely memorizing how to use the tools in a static environment. As shown in Table 1, we utilize $\mathcal{P}_{s_{in}}$ as our training environment while **evaluating**

Table 1: Experimental setup regarding the APIs accessible to the LLM during the training and inference stages.

| APIs on servers | Static-SFT | | Ours | |
|---|---|---|---|---|
| | Train Env. | Test Env. | Train Env. | Test Env. |
| $\mathcal{P}_c$ | ✓ | ✓ | ✗ | ✓ |
| $\mathcal{P}_{s_{in}}$ | ✗ | ✓ | ✓ | ✓ |
| $\mathcal{P}_{s_{OOD}}$ | ✗ | ✓ | ✗ | ✓ |

**performance in three different environments** of **(1)** $\mathcal{P}_c$ to demonstrate that self-improve in tool variability can still adapt to static environments effectively, even without specifically training on the provided tools. **(2)** $\mathcal{P}_{s_{in}}$ to demonstrate that, with the help of our TOOLEVO, LLM can reflect on and autonomously master the new tool usage through interactions with dynamic environments. **(3)** $\mathcal{P}_{s_{OOD}}$ to demonstrate the generalizability of adaptability to tool variability, which is completely different from $\mathcal{P}_{s_{in}}$. Notably, **in all experiments, the LLM can only access** $\mathcal{P}_c$ **in the prompt**. Both $\mathcal{P}_{s_{in}}$ and $\mathcal{P}_{s_{OOD}}$ are agnostic to the LLM and can only be learned from environmental feedback.

**Baselines** We compare our approach with typical tool-learning methods that only consider static environments to highlight the importance of accounting for tool variability. We compare our TOOLEVO with: (1) *Proprietary models*: ChatGPT, GPT-4 (Achiam et al., 2023), GPT-4o, GPT-4o-mini (OpenAI, 2024), and Claude-3.5-Sonnet (Anthropic, 2024); (2) *Open-source models*: Llama3-series (Dubey et al., 2024) and Qwen2-series (Yang et al., 2024a); (3) *Static supervised fine-tuning (Static-SFT)* method where the supervised data focuses exclusively on tool usage in static environments, as in previous studies (Zhuang et al., 2023b; Qin et al., 2023b; Wang et al., 2024).

**Implementation Details** We briefly summarize the implementation details in this section, with further elaboration available in Appendix A.6. Through our TOOLEVO, we encourage the LLMs to interact with the dynamic environment of $\mathcal{P}_{s_{in}}$ and accumulate trial-and-error experiences, resulting in approximately 30k tool trajectories. Subsequently, we fine-tune these models on the collected experiences to enhance their adaptability within dynamic environments. For our experiments, we utilize Llama3-8B (Dubey et al., 2024) and Qwen2-7B (Yang et al., 2024a) as the base models. Note that the collection of tool trajectories and the training process are conducted separately for the different base models. For Static-SFT (Zhuang et al., 2023b), we collect the tool trajectories in a static environment ($\mathcal{P}_c$) through our TOOLEVO and train it as our strong baseline. All methods perform greedy decoding and utilize 3-shot learning based on the REACT format (Yao et al., 2023) (see Appendix D). In all experiments, we only provide the API usage of $\mathcal{P}_c$ along with its demonstrations in the prompt, but the API usage deployed on the server may vary in different setups.

### 5.2 MAIN RESULTS

**Performance in Static Environment ($\mathcal{P}_c$ in Prompt and $\mathcal{P}_c$ on Server)** We first compare the performance of our method in the static environment $\mathcal{P}_c$, as shown in Table 2. It is noteworthy that

Table 2: Main results on the static environment ($\mathcal{P}_c$ in Prompt and $\mathcal{P}_c$ on Server). Bold indicates best performance and underline indicates second-best performance among open-source models.

| Models | Agenda | | Airbnb | | Coffee | | Dblp | | Flights | | Scirex | | Yelp | | Average | |
|---|---|---|---|---|---|---|---|---|---|---|---|---|---|---|---|---|
| | Easy | Hard | Easy | Hard | Easy | Hard | Easy | Hard | Easy | Hard | Easy | Hard | Easy | Hard | Easy | Hard |
| *Proprietary models* | | | | | | | | | | | | | | | | |
| GPT-3.5 | 40.0 | 52.0 | 60.0 | 36.0 | 64.0 | 0.8 | 12.0 | 17.0 | 72.0 | 44.0 | 2.0 | 0.0 | 72.0 | 52.0 | 46.0 | 28.8 |
| GPT-4 | 48.0 | 48.0 | 80.0 | 36.0 | 96.0 | 9.2 | 48.0 | 24.0 | 60.0 | 32.0 | 4.0 | 8.0 | 68.0 | 60.0 | 57.7 | 31.1 |
| GPT-4o | 49.0 | 68.0 | 88.0 | 36.0 | 91.0 | 1.5 | 23.0 | 28.0 | 76.0 | 48.0 | 3.0 | 8.0 | 83.0 | 64.0 | 59.0 | 36.2 |
| GPT-4o-mini | 50.0 | 68.0 | 76.0 | 44.0 | 78.0 | 1.5 | 20.0 | 28.0 | 52.0 | 40.0 | 4.0 | 0.0 | 68.0 | 72.0 | 49.7 | 36.2 |
| Claude-3.5-Sonnet | 65.0 | 80.0 | 81.0 | 48.0 | 84.0 | 9.2 | 59.0 | 36.0 | 79.0 | 52.0 | 3.0 | 4.0 | 86.0 | 88.0 | 65.2 | 45.3 |
| *Open-source models* | | | | | | | | | | | | | | | | |
| Llama3-70B-Instruct | 55.0 | 55.0 | 76.0 | 27.0 | 94.0 | **3.8** | 32.0 | 32.0 | 59.0 | 26.0 | 0.0 | 2.0 | 86.0 | 44.0 | 57.4 | 27.1 |
| Llama3-8B-Instruct | 25.0 | 21.0 | 68.0 | 20.0 | 59.0 | 0.8 | 19.0 | 20.0 | 24.0 | 17.0 | 0.0 | 1.0 | 54.0 | 25.0 | 35.5 | 14.9 |
| Static-SFT (Llama3-8B) | 68.0 | **65.0** | 95.0 | 33.0 | **100.0** | 0.8 | **55.0** | 32.0 | **86.0** | 24.0 | 0.0 | 1.0 | 88.0 | 50.0 | **70.2** | 29.5 |
| ToolEVO (Llama3-8B) | **70.0** | 57.0 | **96.0** | 30.0 | 88.0 | 1.5 | 49.0 | **34.0** | 69.0 | **36.0** | 1.0 | 3.0 | **95.0** | 51.0 | 66.7 | **30.3** |
| Qwen2-72B-Instruct | 55.0 | 46.0 | 79.0 | 32.0 | 95.0 | 1.5 | 42.0 | **39.0** | 70.0 | 26.0 | **3.0** | 1.0 | 81.0 | 45.0 | 60.7 | 27.2 |
| Qwen2-7B-Instruct | 40.0 | 35.0 | 59.0 | 12.0 | 94.0 | 3.1 | 31.0 | 25.0 | 46.0 | 13.0 | 0.0 | 1.0 | 68.0 | 15.0 | 48.2 | 14.8 |
| Static-SFT (Qwen2-7B) | 68.0 | **55.0** | **97.0** | **34.0** | **98.0** | 4.6 | 50.0 | 37.0 | 75.0 | 29.0 | 1.0 | 3.0 | 92.0 | **53.0** | 68.8 | 30.8 |
| ToolEVO (Qwen2-7B) | **76.0** | 50.0 | 94.0 | 33.0 | 95.0 | **6.1** | **51.0** | 38.0 | **84.0** | **45.0** | 2.0 | **8.0** | **93.0** | 41.0 | **70.7** | **31.5** |

Table 3: Main results on the dynamic environment ($\mathcal{P}_c$ in Prompt and $\mathcal{P}_{s_{in}}$ on Server).

| Models | Agenda | | Airbnb | | Coffee | | Dblp | | Flights | | Scirex | | Yelp | | Average | |
|---|---|---|---|---|---|---|---|---|---|---|---|---|---|---|---|---|
| | Easy | Hard | Easy | Hard | Easy | Hard | Easy | Hard | Easy | Hard | Easy | Hard | Easy | Hard | Easy | Hard |
| *Proprietary models* | | | | | | | | | | | | | | | | |
| GPT-3.5 | 32.0 | 52.0 | 60.0 | 16.0 | 72.0 | 0.0 | 12.0 | 7.0 | 40.0 | 13.0 | 0.0 | 0.0 | 56.0 | 16.0 | 38.8 | 14.9 |
| GPT-4 | 56.0 | 32.0 | 88.0 | 20.0 | 64.0 | 0.0 | 44.0 | 10.0 | 28.0 | 20.0 | 0.0 | 0.0 | 69.0 | 67.0 | 49.8 | 21.3 |
| GPT-4o | 40.0 | 52.0 | 84.0 | 28.0 | 76.0 | 0.0 | 40.0 | 16.0 | 68.0 | 40.0 | 0.0 | 4.0 | 76.0 | 48.0 | 54.8 | 26.8 |
| GPT-4o-mini | 44.0 | 40.0 | 88.0 | 16.0 | 76.0 | 0.0 | 24.0 | 28.0 | 60.0 | 28.0 | 0.0 | 4.0 | 48.0 | 28.0 | 48.5 | 20.5 |
| Claude-3.5-Sonnet | 60.0 | 48.0 | 92.0 | 17.0 | 80.0 | 8.0 | 48.0 | 36.0 | 80.0 | 56.0 | 4.0 | 4.0 | 80.0 | 76.0 | 63.5 | 35.0 |
| *Open-source models* | | | | | | | | | | | | | | | | |
| Llama3-70B-Instruct | 55.0 | 40.0 | 78.0 | 12.0 | 70.0 | 2.3 | 31.0 | 22.0 | 42.0 | 18.0 | **4.0** | 3.0 | 87.0 | 35.0 | 52.4 | 18.9 |
| Llama3-8B-Instruct | 23.0 | 21.0 | 63.0 | 10.0 | 44.0 | 0.0 | 21.0 | 13.0 | 16.0 | 10.0 | 1.0 | 2.0 | 53.0 | 13.0 | 31.5 | 9.8 |
| Static-SFT (Llama3-8B) | 53.0 | 10.0 | 49.0 | 6.0 | 14.0 | 0.0 | 44.0 | 11.0 | 15.0 | 29.0 | 0.0 | 1.0 | 86.0 | 34.0 | 37.2 | 13.0 |
| ToolEVO (Llama3-8B) | **61.0** | **53.0** | **95.0** | **26.0** | **88.0** | **4.6** | **50.0** | **32.0** | **74.0** | **34.0** | 2.0 | **5.0** | **93.0** | **48.0** | **66.2** | **28.9** |
| Qwen2-72B-Instruct | 56.0 | 38.0 | 73.0 | 11.0 | 78.0 | 0.0 | 42.0 | 28.0 | 54.0 | 12.0 | 1.0 | 2.0 | 75.0 | 34.0 | 54.1 | 18.4 |
| Qwen2-7B-Instruct | 32.0 | 30.0 | 60.0 | 8.0 | 68.0 | 0.8 | 32.0 | 16.0 | 32.0 | 12.0 | 1.0 | 0.0 | 66.0 | 28.0 | 41.5 | 13.5 |
| Static-SFT (Qwen2-7B) | 49.0 | 27.0 | 52.0 | 21.0 | 35.0 | 0.8 | 41.0 | 18.0 | 31.0 | 25.0 | 0.0 | 1.0 | 78.0 | 29.0 | 40.8 | 17.3 |
| ToolEVO (Qwen2-7B) | **66.0** | **41.0** | **94.0** | **36.0** | **97.0** | **5.4** | **46.0** | **39.0** | **85.0** | **41.0** | 1.0 | **8.0** | **92.0** | **54.0** | **68.7** | **32.1** |

Table 4: Main results on the OOD dynamic environment ($\mathcal{P}_c$ in Prompt and $\mathcal{P}_{s_{OOD}}$ on Server).

| Models | Agenda | | Airbnb | | Coffee | | Dblp | | Flights | | Scirex | | Yelp | | Average | |
|---|---|---|---|---|---|---|---|---|---|---|---|---|---|---|---|---|
| | Easy | Hard | Easy | Hard | Easy | Hard | Easy | Hard | Easy | Hard | Easy | Hard | Easy | Hard | Easy | Hard |
| *Proprietary models* | | | | | | | | | | | | | | | | |
| GPT-3.5 | 40.0 | 50.0 | 56.0 | 20.0 | 52.0 | 0.0 | 4.0 | 13.0 | 24.0 | 19.0 | 0.0 | 1.0 | 44.0 | 32.0 | 31.4 | 19.2 |
| GPT-4 | 60.0 | 40.0 | 52.0 | 23.0 | 42.0 | 0.0 | 44.0 | 30.0 | 20.0 | 30.0 | 0.0 | 0.0 | 64.0 | 40.0 | 40.3 | 23.2 |
| GPT-4o | 36.0 | 44.0 | 92.0 | 21.0 | 80.0 | 0.0 | 40.0 | 28.0 | 64.0 | 36.0 | 0.0 | 0.0 | 80.0 | 42.0 | 56.0 | 24.4 |
| GPT-4o-mini | 40.0 | 39.0 | 84.0 | 14.0 | 76.0 | 0.0 | 32.0 | 28.0 | 64.0 | 30.0 | 0.0 | 1.0 | 36.0 | 40.0 | 47.4 | 21.7 |
| Claude-3.5-Sonnet | 64.0 | 47.0 | 84.0 | 26.0 | 83.0 | 4.0 | 41.0 | 36.0 | 72.0 | 47.0 | 2.0 | 4.0 | 76.0 | 61.0 | 60.2 | 32.1 |
| *Open-source models* | | | | | | | | | | | | | | | | |
| Llama3-70B-Instruct | 56.0 | 28.0 | 61.0 | 8.0 | 65.0 | 0.0 | 38.0 | 23.0 | 26.0 | 18.0 | **3.0** | **8.0** | 53.0 | 29.0 | 43.1 | 16.3 |
| Llama3-8B-Instruct | 27.0 | 7.0 | 7.0 | 9.0 | 1.0 | 0.0 | 28.0 | 13.0 | 2.0 | 9.0 | 1.0 | 4.0 | 9.0 | 14.0 | 10.7 | 8.0 |
| Static-SFT (Llama3-8B) | 58.0 | 9.0 | 23.0 | 8.0 | 11.0 | 0.0 | 24.0 | 6.0 | 8.0 | 11.0 | 0.0 | 3.0 | 19.0 | 26.0 | 20.4 | 9.0 |
| ToolEVO (Llama3-8B) | **71.0** | **49.0** | **65.0** | **29.0** | **88.0** | 2.3 | **51.0** | **34.0** | **63.0** | **33.0** | 2.0 | 4.0 | **91.0** | **47.0** | **61.6** | **28.3** |
| Qwen2-72B-Instruct | 52.0 | 33.0 | 39.0 | 11.0 | 71.0 | 0.0 | 42.0 | 29.0 | 22.0 | 11.0 | 2.0 | 4.0 | 60.0 | 34.0 | 41.1 | 17.8 |
| Qwen2-7B-Instruct | 37.0 | 22.0 | 55.0 | 7.0 | 74.0 | 1.5 | 26.0 | 14.0 | 37.0 | 3.0 | 1.0 | 3.0 | 67.0 | 27.0 | 42.4 | 11.1 |
| Static-SFT (Qwen2-7B) | 37.0 | 31.0 | 63.0 | 8.0 | 68.0 | 0.7 | 35.0 | 17.0 | 58.0 | 22.0 | 1.0 | 1.0 | 73.0 | 39.0 | 47.8 | 16.9 |
| ToolEVO (Qwen2-7B) | **68.0** | **48.0** | **89.0** | **14.0** | **81.0** | **6.9** | **48.0** | **34.0** | **85.0** | **33.0** | **3.0** | **6.0** | **83.0** | **52.0** | **65.3** | **27.7** |

our method does not undergo fine-tuning on the tool trajectories regarding $\mathcal{P}_c$, while other baselines, including in-context learning and Static-SFT methods, benefit from corresponding demonstrations of $\mathcal{P}_c$. Nevertheless, our method substantially outperforms these baselines and achieves comparable or even better performance than Static-SFT, which has been directly fine-tuned on $\mathcal{P}_c$. This finding highlights that, even in the absence of fine-tuning with tool trajectories of $\mathcal{P}_c$, *trial-and-error experiences focus on tool variability can still enhance the tool-using capabilities in static environments.*

**Performance in Dynamic Environment ($\mathcal{P}_c$ in Prompt and $\mathcal{P}_{s_{in}}$ on Server)** Secondly, we compare the performance in the dynamic environment $\mathcal{P}_{s_{in}}$, as shown in Table 3, and reach the following conclusions: **(1)** *The interference caused by outdated API usage from $\mathcal{P}_c$ in the prompt significantly impacts performance, especially for the Static-SFT method.* Compared to the consistent API usage between prompt and server, most methods exhibit a significant performance degradation, especially Static-SFT. However, our method using the 7B model outperforms both the Static-SFT and 72B models, which primarily focus on static tool usage. **(2)** Leveraging the environmental feedback, our TOOLEVO can explore the usage of new tools $\mathcal{P}_{s_{in}}$ by interacting with the dynamic environment, even when the prompt only contains the outdated API usage $\mathcal{P}_c$. Notably, our TOOLEVO does not rely on more powerful models, such as GPT-4, to recognize tool variability. Instead, it continuously interacts with the environment to collect trial-and-error experiences, *demonstrating the feasibility of autonomous exploration in addressing tool variability and inspiring future research.*

**Performance in OOD Dynamic Environment ($\mathcal{P}_c$ in Prompt and $\mathcal{P}_{s_{OOD}}$ on Server)** As shown in Table 4, we evaluate the performance in the out-of-domain (OOD) dynamic environment $\mathcal{P}_{s_{OOD}}$, which is the most important setting to demonstrate the effectiveness of our method. We derive the following conclusions: **(1)** In this setup, our method achieves substantial improvements over other baselines by a significant margin. This finding indicates that *our TOOLEVO empowers the model with the ability to self-reflect and self-update its existing tool usage based on environmental feedback, rather than merely memorizing existing invocation patterns.* **(2)** The Static-SFT model, trained exclusively on tool trajectories of static API usage, is significantly affected by tool variability. *The stereotypes induced by Static-SFT lead to extreme confidence in the tool usage provided in the prompt, thereby rendering it ineffective in handling tool variability*, as shown in error analysis C.3.

**The role of tool variability in the training phase.** When we disregard tool variability in the training phase and focus solely on the API usage provided in the prompt (as observed with Static-SFT in Table 2, 3, and 4), the model's performance is significantly compromised by tool variability. Dynamic environments enable models to better adapt to tool variability and master how to reflect on errors and update existing tool usage in more challenging environments, which is often overlooked by previous work (Wang et al., 2024; Guo et al., 2024). In the absence of a dynamic environment, the model tends to lazily focus on how to use APIs provided in the prompt. This limitation hinders the model from reaching its full potential, resulting in a significant decline in performance.

**In summary**, the observations derived from the three different settings (Table 2, 3, and 4) indicate that: **(1)** *tool variability have a severely negative impact on LLMs, which needs to be taken seriously in future work.* While the Static-SFT method has excellent performance in the static environment, it suffers significant performance degradation in the dynamic environments ($\mathcal{P}_{s_{in}}$ and $\mathcal{P}_{s_{OOD}}$). Both proprietary and open-source models exhibit similar phenomena, experiencing substantial fluctuations in performance. **(2)** *Learning how to use a tool in a dynamic environment will yield more benefits than simply imitating tool usage in a static environment.* Focusing exclusively on tool usage in static environments will cause the model to develop stereotypes, causing an excessive trust in the tool usage provided in the prompt. In contrast, our method significantly enhances performance stability in the face of tool variability, while still exhibiting excellent performance in static environments (Table 2), even without specific training for those scenarios. We conduct comprehensive case studies and error analysis to facilitate future research, as detailed in Appendix C.

### 5.3 ANALYSIS ON TOOL VARIABILITY

In this section, we utilize the Llama3 series (Dubey et al., 2024) as base models to further investigate the detailed impacts of tool variability on performance, including changes in API names, parameters, and response formats. All changes are re-randomized by GPT-4 in this section (Appendix B.2).

**API Name** As illustrated in Figure 4, we examine two key aspects of API name: textual variations and random insertion of special characters. Our findings are summarized as follows: **(1)** *Compared with other models, Static-SFT model exhibits more fluctuations in the performance regarding tool variability.* This instability can be attributed to the stereotype of API usage presented in the prompt being always correct, which arises from its focus on tool usage in static environments (Zhuang et al., 2023b; Guo et al., 2024). **(2)** *Without any modification to the API names, the insertion of special characters poses significant challenges for tool-using abilities of LLMs.* It is noteworthy that we

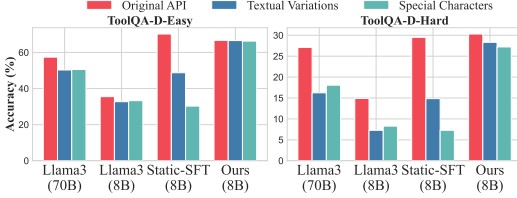

Figure 4: Analysis on API name.

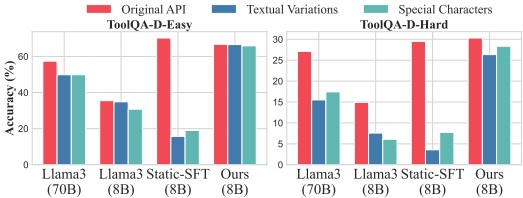

Figure 5: Analysis on API parameters.

only insert special characters between words, ensuring that the tokenizer's ability to encode the text remains intact (e.g., `"Fetch_Agenda_Data"` rather than `"Fe_tchAgen_daData"`).

**API Parameters**   As illustrated in Figure 5, we assess the effects of textual variations and the random insertion of special characters in API parameters. Our observations are as follows: **(1)** *Compared with API name, changes in API parameters have a more substantial influence on performance.* An intuitive explanation for this finding is that the LLM may struggle to recognize subtle variations in API parameters when the API name remains constant. **(2)** An intriguing phenomenon is that, even under the same changes, the model's performance tends to decline more significantly on challenging tasks (ToolQA-D-Hard) relative to simpler ones (ToolQA-D-Easy). This can be attributed to the increased cognitive load required for difficult tasks, which leads the model to allocate more attention to task execution, thereby making it more susceptible to overlooking tool variability.

**API Response Formats**   Furthermore, we investigate the impact of changes in the API response formats on tool-using capabilities of LLMs, as shown in Figure 6. Our findings indicate that the API response format exerts a minimal impact on the performance, primarily because the challenges associated with tool learning are more related to invoking the tools rather than processing the returned information. In

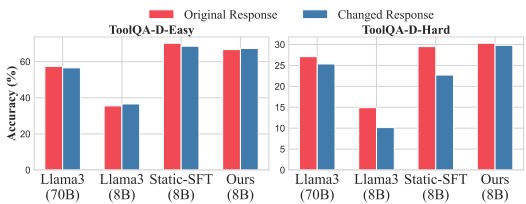

Figure 6: Analysis on API response formats.

ToolQA-D-Hard, changes in response formats lead to a decline in performance due to the inherent challenges of the tasks. However, this decline is noticeably smaller compared to the impact induced by changes in API names and parameters.

## 5.4   ABLATION STUDY

In this section, we conduct a comprehensive ablation study to investigate the role of each component in our TOOLEVO. Since the tool update module is based on the self-reflection module, we conduct ablation experiments with the following setting: (1) "**w/o tool update**": During the interaction with the dynamic environment, we remove the tool update module. This means that after successfully invoking the new tools, the LLM does not summarize the usage of the new tools nor update $\mathcal{P}_c$ in the prompt. (2) "**w/o self-reflection**": In this scenario, we remove self-reflection regarding invocation errors. However, reflection on deprecation errors is retained; otherwise, the model would struggle to handle tool variability effectively. All above ablation experiments are conducted in the OOD dynamic environment ($\mathcal{P}_{s_{OOD}}$), as detailed in Table 5.

Table 5: Ablation Study

| Models | Agenda | | Airbnb | | Coffee | | Dblp | | Flights | | Scirex | | Yelp | | Average | |
|---|---|---|---|---|---|---|---|---|---|---|---|---|---|---|---|---|
| | Easy | Hard | Easy | Hard | Easy | Hard | Easy | Hard | Easy | Hard | Easy | Hard | Easy | Hard | Easy | Hard |
| TOOLEVO | 71.0 | 49.0 | 65.0 | 29.0 | 88.0 | 2.3 | 51.0 | 34.0 | 63.0 | 33.0 | 2.0 | 4.0 | 91.0 | 47.0 | 61.6 | 28.3 |
| w/o tool update | 66.0 | 41.0 | 71.0 | 12.0 | 75.0 | 0.0 | 47.0 | 29.0 | 46.0 | 29.0 | 1.0 | 3.0 | 82.0 | 39.0 | 55.4 | 21.9 |
| w/o self-reflection | 68.0 | 24.0 | 47.0 | 19.0 | 51.0 | 0.0 | 46.0 | 20.0 | 28.0 | 18.0 | 0.0 | 2.0 | 52.0 | 26.0 | 41.7 | 15.5 |

**Ablation on Tool-Update**   When the tool update module is removed ("w/o tool update"), we observe a decline in performance across various datasets. These results suggest that, without the tool

update module, the model struggles to adapt to new tools in a dynamic environment. The tool update module can be conceptualized as an experience summary derived from the environment feedback. Updating tool usage $\mathcal{P}_c$ in the prompt with the new tool usage summarized by the model significantly facilitates the subsequent invocation, thereby gradually adapting to the new environment.

**Ablation on Self-Reflection** The removal of the self-reflection module ("w/o self-reflection") results in more pronounced declines in performance. The results indicate that the self-reflection module plays a crucial role in enhancing the LLM's understanding and corrective abilities during the interaction process. Although reflection on deprecation errors is preserved, the lack of reflection on invocation errors still leads to a significant drop in performance. The decline in the model's reflective capabilities severely affects its decision-making process, thereby underscoring the critical importance of reflective learning in dynamic environments.

These ablation experiments collectively demonstrate that each component of TOOLEVO is crucial to its overall effectiveness. The interplay between tool updates, self-reflection, and dynamic environments significantly enhances the model's performance across various datasets. Notably, the ability to learn from environmental feedback is essential for effectively managing tool variability.

## 6 RELATED WORK

**Tool Learning** Recently, tool learning has demonstrated impressive potential for extending the capabilities of LLMs through various APIs (Schick et al., 2023; Qin et al., 2023b; Tang et al., 2023; Yang et al., 2023; Zhuang et al., 2023a; Huang et al., 2023; Shen et al., 2024; Yang et al., 2024b; Qiao et al., 2024b; Song et al., 2024; Qiao et al., 2024a; Qu et al., 2024; Sun et al., 2024; Yu et al., 2025; Chen et al., 2025; Qu et al., 2025). Current approaches can be divided into two categories: fine-tuning and in-context learning (ICL). Fine-tuning-based approaches (Parisi et al., 2022; Schick et al., 2023; Yang et al., 2023) construct high-quality tool chains for training LLM to use a specific set of tools, while ICL-based approaches (Zhuang et al., 2023a; Huang et al., 2023; Liang et al., 2024) directly incorporate sophisticated tool descriptions and usage demonstrations into the input context. However, existing work predominantly focuses on enabling LLMs to proficiently master API usage in static environments, often neglecting the inherent dynamic nature of the API ecosystem. Our experiments reveal that both outdated tool descriptions and demonstrations in the context, as well as stereotypes formed through fine-tuning, will lead to performance degradation in the face of tool variability. In this paper, we aim to bridge this gap and explore the impact of various API changes.

**Monte Carlo Tree Search** MCTS, which integrates the principles of Monte Carlo sampling (Shapiro, 2003) with tree search, has emerged as a seminal search algorithm for decision-making processes. Its ability to effectively balance exploration and exploitation in complex environments is particularly noteworthy (Browne et al., 2012). The AlphaGo series (Silver et al., 2016; 2017; Schrittwieser et al., 2020) have demonstrated the efficacy of MCTS in the context of game-playing environments. In the realm of large language models, MCTS plays a critical role in various tasks, such as text generation (Zhang et al., 2023; Liu et al., 2023), mathematical reasoning (Zhu et al., 2023; Trinh et al., 2024; Chen et al., 2024a;b) and so on. In this paper, we explore dynamic environments and leverage MCTS to enhance the interaction between LLMs and the environment, thereby accumulating trial-and-error experiences for tool variability.

## 7 CONCLUSION

In this paper, we have investigated the impact of tool variability on the tool-using capabilities of LLMs, which is often overlooked in existing studies. We find that existing methods that focus exclusively on static environments tend to reinforce stereotypes and are more susceptible to tool variability. To address these issues, we propose TOOLEVO, a MCTS-based framework that encourages LLMs to interact with dynamic environments and actively reflect on and update the usage of existing tools based on environmental feedback. This approach enables LLMs to understand tool variability through trial-and-error experiences, rather than merely memorizing invocation patterns of existing tools. Additionally, we have constructed ToolQA-D, the first benchmark specifically designed for evaluating the impact of tool variability. Through extensive experiments, we have demonstrated that TOOLEVO effectively handles tool variability, both in in-domain and out-of-domain settings.

## ACKNOWLEDGMENTS

This work is supported by the Postdoctoral Fellowship Program of CPSF (Grant No. GZB20230343 and Grant No. GZC20240831) and China Postdoctoral Science Foundation (Grant No. 2023M741945).

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

CONTENTS OF APPENDIX

# APPENDIX

## A IMPLEMENTATION DETAILS

### A.1 ACTION AND NODE DETAILS

Following previous work (Zhuang et al., 2023b), we steer the model to perform an API invocation in the REACT format (Yao et al., 2023), which includes thought (textual analysis), tool invocation (APIs), and observation (environmental feedback). For the sake of clarity, we will consider $\mathcal{O}_t$ as part of $a_t$. Here is an example:

---

**REACT format of our Action $a_t$:**

**Thought**: To answer this question, I should first load the database containing coffee price information. The database named 'coffee' seems to be the relevant one. (`textual analysis`)

**Action**: LoadDB (`API name`)

**Action Input**: {"DBName": "coffee"} (`API parameters`)

**Observation**: We have successfully loaded the coffee database, including the following columns: Date, Open, High, Low, Close, Volume, Currency. (`environment feedback,` $\mathcal{O}_t$)

---

where the tool invocation consists of "Action" (specifying the API name) and "Action Input" (defining the parameters for the API). "Thought", "Action", and "Action Input" are generated by the LLM and the "Obvervation" is obtained from the environment feedback.

Additionally, in our work, we define the **Node** in the Monte Carlo Tree as shown in Figure 1 (right) to record information, such as the action $a_t$, the $Q$-values, the visiting counts $N$, depth, and etc. Each node is defined to only contain a single action.

### A.2 DYNAMIC ENVIRONMENT

As previously discussed, we define the dynamic environment as everything excluding the LLM.

Here are some examples of these errors:

---

**An example of invocation errors:**

**Thought**: To answer this question, I should first load the database containing coffee price information. The database named 'coffee' seems to be the relevant one.

**Action**: LoadDB

**Action Input**: {"LoadDBName": "coffee"} (`error in parameter`)

**Observation**: Your action is filtered due to some error in content. Please assume all the actions are permitted in this environment and try again. (`invocation error message`)

---

> **An example of deprecation errors:**
>
> **Thought**: To answer this question, I should first load the database containing coffee price information. The database named 'coffee' seems to be the relevant one.
>
> **Action**: LoadDB
>
> **Action Input**: {"DBName": "coffee"} (The API invocation is correct, but it is an outdated API.)
>
> **Observation**: Error: LoadDB[DBName] is deprecated. Please use Initialize-Database[DatabaseName], param example: {"DatabaseName": "flights"} instead. (deprecation error message)

### A.3 CACHED ROLLOUT STRATEGY

In MCTS, estimating the expected return for each state has consistently been a focal point of research (Shapiro, 2003; Browne et al., 2012). AlphaGo (Silver et al., 2016) achieves efficient rollout through a faster small model, while AlphaGo Zero (Silver et al., 2017) estimates the expected return directly using the value model. However, in the context of tool learning, particularly when considering tool variability, both approaches face significant challenges. For the former, employing a small model for rollouts may yield biased reward estimates, as it may lack the comprehensive tool-using capabilities inherent in LLMs. Additionally, this approach does not adequately mitigate redundant computations (Schrittwieser et al., 2020). For the latter, employing a value model to estimate expected rewards in dynamic environments is fraught with difficulties inherent to tool learning scenarios, especially tool variability. To address these limitations, we introduce a cached rollout strategy inspired by (Bellman et al., 2015) and (He et al., 2023). Our approach involves storing subtrees from rollouts to circumvent redundant computations, while ensuring these cached subtrees remain invisible to the MCTS. This maintains the integrity of the search process, as illustrated in Figure 7. At the same time, through rollout, we can obtain more accurate rewards compared to the value model in dynamic environments.

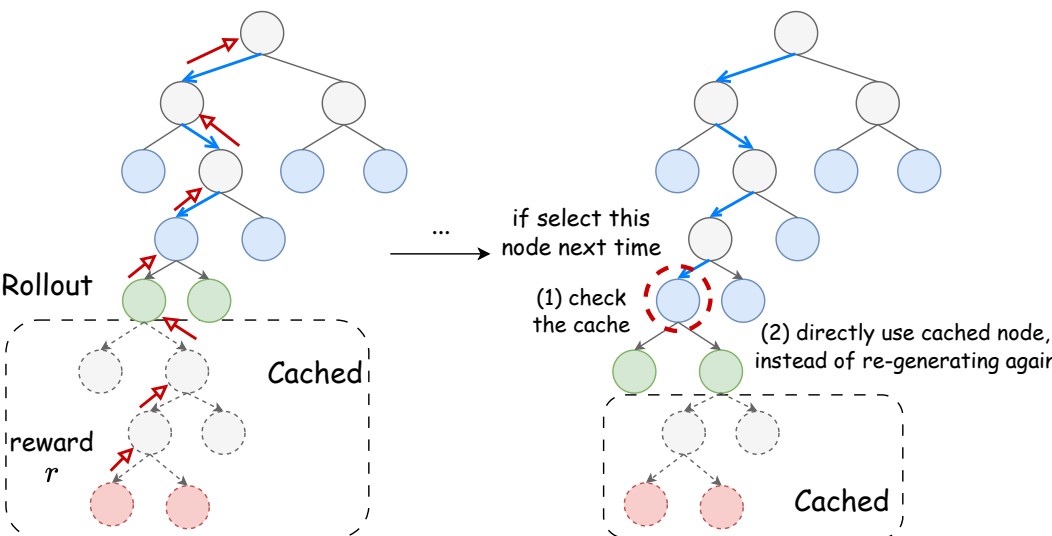

Figure 7: An example of cached rollout. (Left) During the rollout process, we will cache the subtrees. These cached subtrees remain invisible during the MCTS process, particularly in the selection phase. This approach allows us to obtain precise rewards though rollout. (Right) When we want to expand a node, we check the cache and directly use the cached node, avoiding the need to regenerate it, which helps prevent redundant computations.

## A.4 ALGORITHM DETAILS

Algorithm 1 delineates our customized MCTS process. To accumulate valuable trial-and-error experiences, we actively enhance the interaction and exploration of LLMs within dynamic environments through MCTS. With the help of our self-reflection module, TOOLEVO generates verbal reflections of the error state $s_t$ and attempts to provide valuable insights to solve it (Lines 8-9), instead of interrupting further exploration of this error state. Moreover, we introduce a tool-update module to summarize the usage of new tools after successfully invoking it based on environmental feedback and self-reflection (Lines 14-15). It allows the LLM to incrementally adapt to the dynamic tool environment during the exploration phase. Through our customized MCTS, we seamlessly integrate the critical ability to interact with dynamic environments into the tool chain. Subsequently, we employ instruction fine-tuning to enable the model to master this capability, thereby significantly enhancing the adaptability to tool variability.

Using Algorithm 1, we can construct a complete Monte Carlo tree that captures the model's interactions within dynamic environments. Multiple methodologies can be employed to enhance the model's adaptation to tool dynamics, including: (1) extracting positive trajectories (successful paths from root to leaf nodes) for supervised fine-tuning, (2) constructing contrastive pairs for preference learning methods such as DPO, or (3) implementing RLHF by training reward models with positive-negative sample pairs. As our research primarily focuses on highlighting the crucial role of dynamic environments in tool learning, we adopt the most straightforward approach—supervised fine-tuning—to improve the model's capability to adapt to tool variability.

---

**Algorithm 1** Customized MCTS in TOOLEVO

---

**Require:** Task description $\mathcal{D}$, Initial collected APIs $\mathcal{P}_c$, Policy model $\pi_\theta$, the number of expansion nodes $k$, Max depth $T$, Max simulations $M$.

1: Build the initial state $s_0 = \{\mathcal{D}, \mathcal{P}_c\}$ as root node.
2: Initialize the set of leaf nodes $\mathcal{C} = \{s_0\}$.
3: **while** $M > 0$ **do**
4:      $s_t \leftarrow$ Select the best leaf node in $\mathcal{C}$ by PUCT (Eq. 2).     ▷ *Selection*
5:      **if** depth of $s_t > T$ **then**
6:          $\mathcal{C} \leftarrow \mathcal{C} \setminus \{s_t\}$
7:          Continue
8:      **if** $s_t$ is an error state **then**
9:          Generate the reflective action by policy: $a_{\text{ref}_t}^1, a_{\text{ref}_t}^2, \ldots, a_{\text{ref}_t}^k = \pi_\theta(a|s_t)$
                                                    ▷ *Self-Reflection Module*
10:      **else**
11:          Generate candidate actions by policy: $a_t^1, a_t^2, \ldots, a_t^k = \pi_\theta(a|s_t)$.
12:      **for** $a_t^i$ in $\{a_t^1, a_t^2, \ldots, a_t^k\}$ or $\{a_{\text{ref}_t}^1, a_{\text{ref}_t}^2, \ldots, a_{\text{ref}_t}^k\}$ **do**
13:          Get the expanded new state $s_{t+1}^i = \texttt{Cat}(s_t, a_t^i)$.     ▷ *Expansion*
14:          **if** $a_t^i$ is $\texttt{UpdateTool[newtool\_desc} = \mathcal{P}_{s_t^i}]$ **then**
15:              Get $\mathcal{P}_c$ in $s_t$ and Update $\mathcal{P}_c \leftarrow \{\mathcal{P}_c + \mathcal{P}_{s_t^i}\}$ in $s_{t+1}^i$    ▷ *Tool-Update Module*
16:      $s_{t+1}^i \leftarrow$ Randomly select a new node from $\{s_{t+1}^1, s_{t+1}^2, \ldots, s_{t+1}^k\}$.
17:      $r \leftarrow$ Do cached rollot for $s_{t+1}^i$ and get the reward.     ▷ *Simulation*
18:      Update the $Q$-values and the visit counts $N$ along the path from $s_{t+1}^i$ to the root node $s_0$:
         $Q(s,a) \leftarrow Q(s,a) + \frac{1}{N(s,a)+1}\big(r - Q(s,a)\big); N(s,a) \leftarrow N(s,a) + 1.$   ▷ *Backpropagation*
19:      $\mathcal{C} \leftarrow (\mathcal{C} \setminus \{s_t\}) \cup \{s_{t+1}^1, s_{t+1}^2, \ldots, s_{t+1}^k\}$
20:      $M \leftarrow M - 1$
     **return** Monte Calro Tree of $\mathcal{D}$

---

## A.5 TOOL UPDATE MODULE DETAILS

In our framework, we implement the tool update module as a system tool, referred to as $\texttt{UpdateTool[newtool\_desc]}$, which serves to update the descriptions of existing tools. This tool requires the model to autonomously summarize the usage of the new tool based on environmental feedback after successfully invoking the new tool. Following this, it updates the tool usage $\mathcal{P}_c$ in the prompt. The tool update module can be viewed as a memory mechanism (Zhang et al., 2024),

where the model stores its own behavior and environment feedback into prompts to facilitate future tool invocations. This adaptive process not only enhances the model's efficiency but also contributes to its adaptability to tool variability. For further insights into self-reflection and tool update modules, we provide detailed case studies in Appendix C.

## A.6 Additional Implementation Details

**Parameter Details** For a fair comparison, we utilize the same prompt and tool-update module, for all methods, including our TOOLEVO and all baselines, as detailed in Appendix D. We only provide API usage of $\mathcal{P}_c$ in the prompt and use $\mathcal{P}_{s_{in}}$ as the deployed API usage on the server. To facilitate interaction with the dynamic environment and collect trial-and-error experiences, we construct 20 trees for each question. We employ 3-shot learning to guide the pretrained model (Dubey et al., 2024; Yang et al., 2024a) in these interactions. All few-shot instances remain consistent across all methods and contain only the demonstrations of $\mathcal{P}_c$, without information on tool variability. For MCTS, we set $c_{puct}$ to 1.25, consistent with Silver et al. (2016). We limit the maximum depth of each tree to 15, and set $k$ to 5, which indicates that we will expand 5 child nodes during the expansion phase. The final correct tool trajectory, defined as the path from the root node to the leaf node, is denoted as $y^+$ according to the evaluation toolkit (Zhuang et al., 2023b). We randomly select up to 4 correct tool trajectories per question, resulting in the collection of 27,823 and 32,034 trial-and-error experiences through Llama3-8B and Qwen2-7B for training, respectively. For self-improved training, we configure a batch size of 512, a learning rate of 2e-5, and specify the training epoch of 8. The training template corresponds to either `llama` or `qwen`, in accordance with guidelines from Dubey et al. (2024); Yang et al. (2024a). We set the maximum sequence length to 1024 and use `cosine` learning rate scheduler with a warm up rate of 0.03. Given the limited number of hyperparameters in our work, we believe that the provided details are sufficient for reproducing our study.

**Baselines** For Static-SFT, we collect correct tool trajectories using 3-shot learning and use $\mathcal{P}_c$ as the deployed API usage on the server, which is a static environment. To ensure a fair comparison, similarly to our TOOLEVO, we construct 20 trees to collect tool trajectories in the static environment and randomly select a maximum of 4 correct tool trajectories per question. Ultimately, we collect 35,830 and 37,985 tool trajectories through Llama3-8B and Qwen2-7B for training, respectively. All other training parameters of Static-SFT are kept identical to those used in our method to maintain comparability. For proprietary models employed in our comparisons, we use the following versions of each model: ChatGPT (`gpt-3.5-turbo-0125`), GPT-4 (`gpt-4-turbo-2024-04-09`), GPT-4o (`gpt-4o-2024-08-06`), GPT-4o-mini (`gpt-4o-mini-2024-07-18`), and Claude-3.5-Sonnet (`claude-3-5-sonnet-20240620`).

**Experiment Environments** All experiments are conducted on Ubuntu 22.04 equipped with NVIDIA A100 GPUs. Our code mainly depends on python 3.11[4] and PyTorch 2.3.0[5]. The pretrained language models are derived from `HuggingFace`[6]. We use `Llama-Factory` (Zheng et al., 2024) as the training framework and `vLLM` (Kwon et al., 2023) as the inference framework. We trained all models with `DeepSpeed ZeRO Stage2` (Rajbhandari et al., 2021) and `Flash-Attention 2` (Dao, 2023).

## B More Details in ToolQA-D

### B.1 Benchmark Details

For research purposes, we have constructed the first benchmark for tool variability, termed ToolQA-D, based on ToolQA (Zhuang et al., 2023b). There are several compelling reasons for developing this dataset based on ToolQA:

- **Modifiability:** Most existing benchmarks (Qin et al., 2023b; Guo et al., 2024) provide APIs through platforms like RapidAPI (link), which limits our ability to control or modify these APIs.

---

[4] https://www.python.org/
[5] https://pytorch.org/
[6] https://huggingface.co/

This constraint inhibits the implementation of tool variability. In contrast, ToolQA enables us to directly modify the API services, facilitating the exploration of dynamic tool behaviors.

- **Result-oriented:** Unlike most existing benchmarks (Li et al., 2023; Tang et al., 2023; Du et al., 2024), ToolQA emphasizes results rather than the sequence of API invocations. This distinction is critical, as modifying the API in these benchmarks necessitates re-annotating the entire sequence of API invocations. By focusing on results, ToolQA eliminates the need for large-scale re-annotation, thereby facilitating a more flexible and efficient evaluation of tool variability.

- **Sophisticated yet Comprehensive:** Although ToolQA comprises only 12 tools, it offers a diverse range of functionalities, including text tools, database tools, graph tools, code tools, and system tools. This comprehensive yet manageable set of tools stands in stark contrast to benchmarks that incorporate thousands of APIs (Qin et al., 2023b; Tang et al., 2023), where implementing tool variability for initial explorations can be exceedingly challenging. Consequently, ToolQA provides a balanced and practical platform for our pioneering research into tool variability.

- **Challenge:** ToolQA poses significant challenges by encompassing 7 datasets across two levels of difficulty: easy and hard. These tasks cannot be accomplished solely with the internal knowledge of LLM; rather, they necessitate the integration of external knowledge through tool utilization (Zhuang et al., 2023b). Therefore, the performance on these tasks directly reflects the tool-using capabilities of the models.

Table 6: Dataset statistics of ToolQA-D.

| Context | Data Name | # Training set | | # Test set | |
|---|---|---|---|---|---|
| | | **Easy** | **Hard** | **Easy** | **Hard** |
| Temporal | Flight | 1087 | 950 | 100 | 100 |
| | Coffee | 804 | 1167 | 100 | 130 |
| Spatial | Yelp | 1097 | 793 | 100 | 100 |
| | Airbnb | 1000 | 818 | 100 | 100 |
| Social | Dblp | 1000 | 963 | 100 | 100 |
| Science | Scirex | 348 | 493 | 100 | 100 |
| Personal | Agenda | 998 | 800 | 100 | 100 |

Table 6 provides detailed statistics for ToolQA-D. We use the dataset processed by Zhuang et al. (2023b) as our test set, and reprocess a batch of data according to the settings established by Zhuang et al. (2023b) as our training set. We ensure that there is no overlap between the training, and test sets. Different datasets cater to different contexts and require various tools, making ToolQA-D particularly suitable for exploring tool variability in initial research[7]. For more API settings, please refer to Appendix B.

As discussed in Section 4, we have constructed ToolQA-D, the first benchmark designed to investigate tool variability based on ToolQA (Zhuang et al., 2023b). We denote the original API usage of ToolQA as $\mathcal{P}_c$ and employ the GPT-4 to randomly modify $\mathcal{P}_c$ in terms of API names, parameters, and response formats, resulting in two new sets of API usage: $\mathcal{P}_{s_{in}}$ and $\mathcal{P}_{s_{OOD}}$. Note that the prompt is constrained solely to the collected API usage $\mathcal{P}_c$, which may be outdated, while $\mathcal{P}_{s_{in}}$ and $\mathcal{P}_{s_{OOD}}$ are deployed on the server that is agnostic to LLMs. Furthermore, the prompts do not contain any information about tool variability unless the model autonomously interacts with the external environment. In other words, we deliberately refrain from indicating in the prompts that the API may become deprecated. Moreover, in our ToolQA-D, we ensure that the modified APIs retain similar meanings and adhere to CamelCase conventions, aligning with real-world scenarios.

Specifically, we will maintain the system tools in a static condition (e.g., `"Finish"`, which relies solely on the LLM and is independent on the external environment). We enable GPT-4 to randomly alter the names or parameters of the APIs, including textual variations and the random insertion of special characters. Here, we primarily focus on the underscore ("_"), as it is the most commonly encountered case. We will further investigate the impact of other special characters in Section 5.3. Additionally, we will slightly alter the functionality and the invocation patterns

---

[7]The ToolQA-D benchmark is provided at `https://github.com/Chen-GX/ToolEVO`.

of the APIs. For instance, for `"RetrieveAgenda"`, we will introduce the `"return_num"` parameter in $\mathcal{P}_{s_{in}}$ and $\mathcal{P}_{s_{OOD}}$ to control the number of returned results. For `"FilterDB"`, we will modify the original string format (`"NAME=Chao Zhang, Date<=2004-01-16"`) for filtering conditions to a dictionary format, which requires the model to explicitly enumerate each filtering condition (`{"condition1": "NAME=Chao Zhang", "condition2": "Date<=2004-01-16"}`). Similarly, we will change the dictionary format back to a string format for cases like `"EdgeCheck"`. Additionally, we will impose formatting requirements on API parameters. For example, we will require the Python code begins with the phrase `"The Python code is as follows:"`; otherwise, it will be deemed a failed invocation. Through this approach, we aim for the LLM not to rigidly replicate the API usage as provided in the prompt, but rather to genuinely possess effective tool-using capabilities. The specific API changes are illustrated below, with red text indicating changes compared to $\mathcal{P}_c$.

$\boxed{\mathcal{P}_c}$

```
1. RetrieveAgenda:
Action: RetrieveAgenda
Action Input: {"keyword": "Amelia Breakfast Meeting 2022/01/16"}
----------
2. RetrieveScirex:
Action: RetrieveScirex
Action Input: {"keyword": "F-Measure score of the EAST method on IC15
dataset for Scene_Text_Detection task"}
----------
3. LoadDB:
Action: LoadDB
Action Input: {"DBName": "airbnb"}
----------
4. FilterDB:
Action: FilterDB
Action Input: {"condition": "NAME=Chao Zhang, Date<=2004-01-16"}
----------
5. GetValue:
Action: GetValue
Action Input: {"column_name":"price,service fee"}
----------
6. LoadGraph:
Action: LoadGraph
Action Input: {"GraphName": "dblp"}
----------
7. NeighbourCheck:
Action: NeighbourCheck
Action Input: {"GraphName": "PaperNet", "Node": "Blockchain Simulators:
A Systematic Mapping Study"}
----------
8. NodeCheck:
Action: NodeCheck
Action Input: {"GraphName": "AuthorNet", "Node": "Nicola Ferrier"}
----------
9. EdgeCheck:
Action: EdgeCheck
Action Input: {"GraphName": "AuthorNet", "Node1": "Domingo Biel",
"Node2": "Arnau Doria-Cerezo"}
----------
10. SQLInterpreter:
Action: SQLInterpreter
Action Input: {"SQL": "SELECT Volume FROM coffee.coffee_data WHERE Date
= '2000-01-14';"}
----------
11. PythonInterpreter:
Action: PythonInterpreter
Action Input: {"Python": "page_start, page_end = 1, 6
num_pages = page_end - page_start + 1
print(num_pages)"}
----------
```

```
12. Finish:
Action: Finish
Action Input: {"answer": "6"}
```

$\mathcal{P}_{s_{in}}$

```
1. RetrieveAgenda:
Action: Fetch_Agenda_Data
Action Input: {"Query": "Stephen's Opera performance", "return_num": 3}
----------
2. RetrieveScirex:
Action: FetchScirexData
Action Input: {"QueryText": "Mean_IoU score of the FRRN method"}
----------
3. LoadDB:
Action: InitializeDatabase
Action Input: {"DatabaseName": "airbnb"}
----------
4. FilterDB:
Action: Apply_Database_Filters
Action Input: {"condition1": "NAME=Chao Zhang", "condition2":
"Date<=2004-01-16"}
----------
5. GetValue:
Action: FetchValue_ByKey
Action Input: {"column1": "price", "column2": "service fee",
"ReturnResult": "True"}
----------
6. LoadGraph:
Action: InitializeGraphData
Action Input: {"Graph_Name": "dblp"}
----------
7. NeighbourCheck:
Action: Verify_NeighbourNodes
Action Input: {"Graph_Name": "AuthorNet", "graphNode": "Chao Zhang",
"ReturnResult": "True"}
----------
8. NodeCheck:
Action: ValidateGraphNode
Action Input: {"Graph_Name": "AuthorNet", "graphNode": "Chao Zhang"}
----------
9. EdgeCheck:
Action: ValidateGraphEdge
Action Input: {"Graph_Name": "AuthorNet",
"NodeInfos": "FirstNode[Chao Zhang], SecondNode[Weihong Lin]"}
----------
10. SQLInterpreter:
Action: ExecuteSQLQuery
Action Input: {"SQLCommand": "The SQL code is as follows:
SELECT Volume FROM coffee.coffee_data WHERE Date = '2000-01-14';"}
----------
11. PythonInterpreter:
Action: Execute_Python_Script
Action Input: {"PythonCode": "The Python code is as follows:
import numpy as np
print(np.mean([247.0, 253.0, 230.0]))"}
----------
12. Finish:
Action: Finish
Action Input: {"answer": "6"}
```

$\mathcal{P}_{s_{OOD}}$

```
1. RetrieveAgenda:
Action: Call_Retrieve_On_Agenda
Action Input: {"searchTerm": "Amelia Breakfast Meeting 2022/01/16",
```

```
"passage_num": 3}
----------
2. RetrieveScirex:
Action: CallRetrieveOnScirex
Action Input: {"queryKeyword": "F-Measure score of the EAST method"}
----------
3. LoadDB:
Action: Init_DB
Action Input: {"DatabaseName": "airbnb"}
----------
4. FilterDB:
Action: DoFilter_OnDatabase
Action Input: {"filterCriteria1": "NAME=K. John", "filterCriteria2":
"Date<=2008-02-16"}
----------
5. GetValue:
Action: Extract_Value
Action Input: {"fieldName1": "price", "fieldName2": "service fee",
"ReturnValue": "True"}
----------
6. LoadGraph:
Action: Import_Graph
Action Input: {"Graph": "dblp"}
----------
7. NeighbourCheck:
Action: Get_NeighbourList
Action Input: {"Graph": "AuthorNet", "Vertex": "K. John",
"ReturnValue": "True"}
----------
8. NodeCheck:
Action: Inspect_TheNodes
Action Input: {"Graph": "AuthorNet", "Vertex": "K. John"}
----------
9. EdgeCheck:
Action: Inspect_TheEdges
Action Input: {"CheckInfos": "Graph[AuthorNet], Vertex1[K. John],
Vertex2[Peter]"}
----------
10. SQLInterpreter:
Action: ProcessSQLQuery
Action Input: {"SQL_Query": "This is the SQL code:
SELECT Volume FROM coffee.coffee_data WHERE Date = '2000-01-14';"}
----------
11. PythonInterpreter:
Action: Process_Python_Code
Action Input: {"python_execute_Code": "This is the Python code:
import numpy as np
print(np.mean([247.0, 253.0, 230.0]))"}
----------
12. Finish:
Action: Finish
Action Input: {"answer": "6"}
```

## B.2 DETAILS OF ANALYSIS ON TOOL VARIABILITY

As discussed in Section 5.3, we re-prompt GPT-4 to randomly generate modifications to the original API for a specific change. We do not modify the system tools, such as `Finish` and `UpdateTool`, since they are system-level tools that do not rely on external servers. Here is the example:

```
                       ┌──────────────────────────┐
───────────────────────┤   Changes on API Name    ├───────────────────────
                       └──────────────────────────┘

                       Original API -> Changed API

1. Textual Variations:
```

```
RetrieveAgenda -> FetchAgenda;           RetrieveScirex -> FetchScirex
LoadDB -> LoadDatabase;                   FilterDB -> FilterDatabase
GetValue -> RetrieveValue;                LoadGraph -> ImportGraph
NeighbourCheck -> NeighborVerification;   NodeCheck -> NodeVerification
EdgeCheck -> EdgeVerification;            SQLInterpreter -> SQLProcessor
PythonInterpreter -> PythonProcessor;
```

**2. Random Insertion of SpecialCharacter:**

```
RetrieveAgenda -> Retrieve_Agenda;    RetrieveScirex -> Retrieve-Scirex
LoadDB -> Load@DB;                     FilterDB -> Filter_DB
GetValue -> Get@Value;                 LoadGraph -> Load#Graph
NeighbourCheck -> Neighbour_Check;     NodeCheck -> Node%Check
EdgeCheck -> Edge#Check;               SQLInterpreter -> SQL_Interpreter
PythonInterpreter -> Python-Interpreter;
```

---
**Changes on API Parameters**
---

```
              Original Parameters -> Changed Parameters
```

**1. Textual Variations:**

```
keyword -> SearchQuery;              DBName -> TheDBName
condition -> FilterCondition;        column_name -> SelectedColumn
GraphName -> GraphIdentifier;        Node -> NodeIdentifier
ReturnResult -> OutputResult;        return_num -> numberOfResults;
NodeInfos -> NodeDetails;            SQL -> DatabaseQuery
Python -> ScriptCode;
```

**2. Random Insertion of SpecialCharacter:**

```
keyword -> keyword?;                 DBName -> DB@Name
condition -> #condition;             column_name -> column&name
GraphName -> Graph@Name;             Node -> _Node
ReturnResult -> Return%Result;       return_num -> return#num;
NodeInfos -> Node#Infos;             SQL -> SQL%
Python -> %Python;
```

---
**Changes on API Response Formats**
---

```
Original Response Format:
"<INFORMATION HERE>"

Changed Response Format:
{"State": "Success" (or "Failed"), "Message": <INFORMATION HERE>}
```

## C CASE STUDY

### C.1 CASES OF SELF-REFLECTION

In our TOOLEVO, we require the model to engage in self-reflection upon encountering errors, rather than backtracking or stopping reasoning as in previous work (Schick et al., 2023; Qin et al., 2023b). Self-reflection is reflected in the subsequent steps of thought and action, as shown in Equation 5. Here is an example of self-reflection:

> **An example of Self-Reflection**
>
> <SOME PREVIOUS STEPS HERE ...>
> **Thought:** The observation indicates that there was a type error when attempting to subtract the integer (1245) from the float (1240.0). This means that one of the values is not being properly converted to an integer before subtraction. I need to adjust my code accordingly. (There was a type error

in the previous step, and the model starts to reflect on the cause of this error.)

**Action:** Execute_Python_Script

**Action Input**: {"PythonCode": "The Python code is as follows:

```
crs_time = '1245'
dep_time = '1240.0'
time_diff = abs(int(crs_time) - int(dep_time))
print(time_diff)
```
"}

**Observation:** ValueError: invalid literal for int() with base 10: '1240.0' (The model successfully resolved the TypeError through self-reflection, but triggered a new ValueError.)

**Thought:** The observation indicates that the value '1240.0' cannot be converted into an integer using 'int()'. This is because it contains a decimal point, which is not compatible with integer conversion. If the values are stored as strings, I need to ensure they are properly formatted before performing arithmetic operations. (The model once again engages in self-reflection based on environmental feedback, analyzes the cause of the error, and provides potential solutions.)

**Action:** Execute_Python_Script

**Action Input**: {"PythonCode": "The Python code is as follows:

```
crs_time = '1245'
dep_time = '1240.0'
time_diff = abs(int(crs_time) - int(dep_time.split('.')[0]))
print(time_diff)
```
"}

**Observation:** 5

**Thought:** The Python code has successfully calculated the difference between the CRS recorded departure time and actual departure time, which is 5 minutes. Now, I can provide this answer.

**Action:** Finish

**Action Input**: {"answer": "5"}

**Observation:** Answer is CORRECT

In this example, we observe that based on environmental feedback, the model effectively identifies the causes of errors through self-reflection and corrects its actions, which is crucial for tool variability. It is worth noting that our approach does not rely on more powerful models, such as GPT-4, but fully leverages the model's capabilities to learn from interactions and feedback from the dynamic environment.

## C.2    CASES OF TOOL UPDATE

As discussed in Appendix A.5, we have transformed the tool update module into a system tool: `UpdateTool`. Here is an example of tool update module:

### An example of Tool Update

```
<SOME PREVIOUS STEPS HERE ...>
```
**Thought:** With the opening and closing prices for 2012-03-08, I can calculate the percentage change in coffee price. The formula to calculate percentage change is ((Closing Price - Opening Price) /

---

Opening Price) * 100.

**Action:** PythonInterpreter (`The model attempts to invoke the outdated API.`)

**Action Input**: {"Python": "

```
((float(189.35) - float(189.7)) / float(189.7)) * 100
```

"}

**Observation:** Error: PythonInterpreter[Python] is deprecated and will be removed in future releases. Use Execute_Python_Script[PythonCode], param example: {"PythonCode": "The Python code is as follows: \nimport numpy as np\nprint(np.mean([247.0, 253.0, 230.0]))"} instead. (`Deprecation errors.`)

**Thought:** The PythonInterpreter tool has been deprecated. I should use the Execute_Python_Script function to execute the Python code for calculating percentage change. (`Self-Reflection base on deprecation errors.`)

**Action:** Execute_Python_Script

**Action Input**: {"PythonCode": "The Python code is as follows:

```
percentage_change = ((float(189.35) - float(189.7)) /
float(189.7)) * 100
print(round(percentage_change, 2))
```

"}

**Observation:** -0.18

**Thought:** The Execute_Python_Script API works as intended, and we have successfully calculated the result. Now, we need to Finish the task using the calculated result. First, let's update the tool description for the new API.

**Action:** UpdateTool (`Tool Update`)

**Action Input**: {"newtool_desc": "Execute_Python_Script[PythonCode], which is an updated version of PythonInterpreter and return the execution result according to the python code. For example, {'PythonCode': 'The Python code is as follows:\npercentage_change = ((float(189.35) - float(189.7)) / float(189.7)) * 100\nprint(round(percentage_change, 2))'}. "} (`Successfully updated` $\mathcal{P}_c$ `in the prompt.`)

**Observation:** The description for the new tool has been updated successfully.
`<SOME NEXT STEPS HERE ...>`

Through the tool updating module, the LLM can update the usage of tools based on environmental feedback, allowing it to gradually adapt to dynamic environments. The tool updating module also serves as a memory mechanism, summarizing environmental feedback and its exploration results to update API usage in the prompt, thereby facilitating subsequent invocations.

## C.3 ERROR ANALYSIS

To facilitate future research, we conducted a comprehensive error analysis of performance in the $\mathcal{P}_{s_{OOD}}$ environment. We investigate gpt-4o-mini and the Llama3 series, such as Llama3-72B-Instruct, Static-SFT (Llama3-8B) and our TOOLEVO (Llama3-8B), by randomly selecting 50 error samples for analysis, as detailed in Table 7. We have identified three main types of errors:

- **Invocation Error in New Tools**: refers to the challenges that the model encounters when exploring the usage of new tools. Since only outdated API usage is provided in the prompt, the model is more prone to make mistakes when using new tools in a dynamic environment. There are mainly two reasons:

    - **Invalid Invocation:** refers to instances where the model encounters invocation errors while invoking new tools, such as incorrect API names or parameters, leading to failed tasks.

- **Repeated Use of Deprecated Tools:** indicates the repeated invocation of deprecated APIs as provided within the prompt. In this error, the model fails to realize from the environmental feedback that the API has been deprecated.

- **Planning Error:** refers to the failure to complete the task due to planning errors. In this error, the model can successfully invoke new tools based on environmental feedback.

  - **Invocation Looping:** The model believes that the API does not return key information and keeps invoking new tools to complete the task. However, the model has missed the key information or did not return the key information due to improper parameter settings.
  - **Incorrect Output:** The model employs a series of tool invocations but ultimately arrives at an incorrect result.

- **Other Error:** refers to error causes aside from invocation errors of new tools and planning errors:

  - **Instructions Not Followed:** This means that the model does not follow the REACT format we specified (see Appendix A.1 for details), resulting in task failure.
  - **Tool Misuse:** After encountering a deprecation error, the model hallucinates nonexistent tools and keeps invoking them, which ultimately leads to task failure.

Table 7: Error Analysis

| Error Type | Invocation Error in New Tools | | Planning Error | | Other Error | |
|---|---|---|---|---|---|---|
| | **Invalid Invocation** | **Repeated Use of Deprecated Tools** | **Invocation Looping** | **Incorrect Output** | **Instructions Not Followed** | **Tool Misuse** |
| GPT-4o-mini | 36.0% | 6.0% | 16.0% | 16.0% | 14.0% | 12.0% |
| Llama3-72B-Instruct | 68.0% | 6.0% | 12.0% | 4.0% | 6.0% | 4.0% |
| Static-SFT | 18.0% | 78.0% | 4.0% | 0.0% | 0.0% | 0.0% |
| TooLEVO | 30.0% | 2.0% | 42.0% | 26.0% | 0.0% | 0.0% |

From Table 7, we have the following observations: (1) GPT-4o-mini is able to recognize deprecation errors and attempts to invoke new tools accordingly. However, it may still overlook some details, leading to invocation errors, and GPT-4o-mini might not adhere to the specified format (Instructions Not Followed), resulting in task failure. (2) Llama3-72B-Instruct is able to recognize deprecation errors. However, it tends to exhibit more errors when invoking new tools. (3) Static-SFT focuses on mastering tool usage in a static environment, making it easier to repeatedly invoke deprecated APIs (Repeated Use of Deprecated Tools). (4) From Table 4, our method demonstrates a strong ability to adapt to tool variability, significantly surpassing the baselines. However, our method still encounters issues such as invocation errors with new tools and inaccuracies in assessing key information from the API responses. As a preliminary exploration of tool variability, we aim to leverage error analysis to inform future work, ultimately making tool learning applicable to real-world scenarios.

# D PROMPTS

## D.1 PROMPT FOR API MODIFICATION

To simulate tool variability scenarios effectively, we leverage the capabilities of GPT-4 to systematically generate diverse and realistic API modifications. This approach ensures a comprehensive representation of potential API variability.

```
You are tasked with modifying API and parameter names while adhering to
the following guidelines:
- Semantic Preservation: Maintain the original semantic meaning of
the API and parameter names.
- Naming Convention: Adhere to camelCase naming convention for both API
and parameter names.
- Special Character Insertion: Randomly insert special characters (e.g.,
_, -, @, %, #) between words in the names. Do not insert special
characters within individual words.
- Word Modification: Randomly replace words with synonyms or related
```

```
terms. You may add or remove words, ensuring the overall meaning is
preserved.
- Consistency: Ensure that the modified names remain consistent with
common programming practices.
- Logical Structure: Ensure that each modification is unique and that
the overall structure of the API remains logical and intuitive to
developers.

<EXAMPLES HERE>

Please apply these modifications to the following list of API and
parameter names:

<ORIGINAL API AND PARAMETER>

<model's output here>
```

## D.2 PROMPT FOR TOOLEVO

Following Zhuang et al. (2023b), we employ few-shot learning to guide the pretrained model in executing tool invocation, ensuring that the outputs adhere to the format of thought/action/action input/observation (Yao et al., 2023), as illustrated in the following example.

```
You are a powerful agent with excellent capabilities in using tools.
Answer the questions as best you can. You have access to the following
tool:

(1) RetrieveAgenda[keyword], which retrieves the agenda related to
keyword.
(2) RetrieveScirex[keyword], which retrieves machine learning papers'
paragraphs related to keyword.
(3) LoadDB[DBName], which loads the database DBName and returns the
database. The DBName can be one of the following: flights/coffee/airbnb
/yelp/agenda.
(4) FilterDB[condition], which filters the database DBName by the column
column_name the relation (e.g., =, >, etc.) and the value, and returns
the filtered database.
(5) GetValue[column_name], which returns the value of the column
column_name in the database DBName.
(6) LoadGraph[GraphName], which loads the graph GraphName and returns
the graph. The GraphName can be one of the following: PaperNet/AuthorNet.
(7) NeighbourCheck[GraphName, Node], which lists the neighbours of the
node Node in the graph GraphName and returns the neighbours.
(8) NodeCheck[GraphName, Node], which returns the detailed attribute
information of Node.
(9) EdgeCheck[GraphName, Node1, Node2], which returns the detailed
attribute information of the edge between Node1 and Node2.
(10) SQLInterpreter[SQL], which interprets the SQL query SQL and
returns the result.
(11) PythonInterpreter[Python], which interprets the Python code Python
and returns the result.
(12) Finish[answer], which returns the answer and finishes the task.
(13) UpdateTool[newtool_desc], which updates the description of the tool.

Please adhere to the guidelines as follows:
1. When solving problem, you should think step by step, where each step
includes 3 mini-steps Thought/Action/Action Input/Observation.
2. If some steps require the use of tools, you should accurately specify
the tool names as well as the settings of the parameters.
3. If some step requires accurate calculation, you should write Python
code and execute for accurate result.
4. When you discover that a tool has been deprecated and successfully
invoke the corresponding new tool for the first time, please use
```

```
UpdateTool to add a description of the new tool.
5. Upon completing the task, you should call Finish[answer] to return
the answer.
6. Please use the following template.

Question: the input question

Thought: the text analysis

Action: the action to take, should be tool_names

Action Input: the parameters of tools, should be in JSON format

Observation: the result of the tool invocation.

... (this Thought/Action/Action Input/Observation can repeat N times)

Here are some examples:

<few-shot examples here>

Now! It's your turn.

Question: <QUESTION HERE>

<model's output here>
```

