# OpenReview forum: "Learning Evolving Tools for Large Language Models"
_ICLR.cc/2025/Conference — ICLR 2025 Poster_

### Official Review · Reviewer_VQDs · 2024-10-31

**Soundness:** 2
**Presentation:** 3
**Contribution:** 3
**Rating:** 6
**Confidence:** 3

**Summary:**

The paper tackles API usage of LLMs in the context of changing API specifications. First, the paper introduces a new dataset “ToolQA-D”, an adaptation of ToolQA with two new versions of API specifications that can be used to simulate a dynamic API change. Second, the paper proposes a method to improve LLM tool usage in such a setting: ToolEVO. They use pre-trained models to interact with the dynamic environment and then fine-tune their model using successful runs from those pretrained models. Their method also is encouraged to self-reflect and has the option to update the API documentation used in the prompt. They evaluate their model empirically on the ToolQA-D dataset and compare it to static approaches (not trained in a dynamic environment) and non-finetuned models. They also provide an ablation study regarding the self-reflection and tool-update modules of their method.

**Strengths:**

The paper tackles an important topic and provides promising results. Their method outperforms the presented baselines and could improve API usage for LLM agents in the future.

**Weaknesses:**

My main concerns are with the presentation and some open questions that i would want to be answered to better judge the results before publication:
- The paper is at times hard to follow, and I found myself constantly looking into the Appendix to be able to follow. Even then some details are still unclear. (See questions)
- The presentation of the results could also be clearer and more convincing. (See questions)

**Questions:**

- How was GPT-4 used to modify the API usage, and how is $P_{S_{ood}}$ completely different from $P_{S_{in}}$? Is the change in API different for every training/test example or is there one change in $P_{S_{ood}}$ and on in $P_{S_{in}}$ (is the variable name change same for all examples in each set)? Is it just the same dataset but with a different change in API parameter names?
- What are the numbers in table 2-4 exactly? Are those the number of successful test cases solved (100 according to the appendix)? If so, why is every problem exactly x.0 except for “Coffee hard”. What is different there that there are decimal points? Also the authors talk about significant margins: Are there any statistical tests performed to support that claim? Are those results even from multiple runs?
- How is the tool-update model used during testing? If the method updates the usage of the API in the first test case correctly, can it use the updated API documentation for the consequent tests? Or is it reset per test instance?
- Does this method also generalize to real out-of-distribution tests? E.g., how would a model be able to adapt to an API that was not included in the training set?

Minor comments:
- Figure 2: Why is there no drop in performance indicator for your method, but for the others?
- It is unclear from the text and captions of table 2-4 what bold and underlined indicates. I would assume that bold is best performance and underline is second best?
- The abbreviation SFT is never explained.

---

> ### Author Response · Authors · 2024-11-15
>
> Dear Reviewer VQDs,
>
> We sincerely thank you for your thorough evaluation and for acknowledging that our paper tackles an important topic and provides promising results. Below, we address each of your comments in detail.
>
> > Weakness
>
> We appreciate your valuable feedback. Due to space constraints, we've been striving to convey the importance of tool variability scenarios for tool learning and our proposed method within the main text. We acknowledge the challenge in balancing comprehensive information with conciseness. Based on your questions, we will revise the main text to ensure a better reading experience. Specific details are addressed in our responses to your questions below.
>
> > Q1
>
> We used GPT-4 to systematically generate diverse and realistic API modifications. The specific prompt is added to the `Appendix D`.
> As shown in `Line 270-277 and Appendix B`, $\mathcal{P_{S_{in}}}$ and $\mathcal{P_{S_{OOD}}}$ are indeed based on the same underlying dataset, addressing the same problem domain. However, $\mathcal{P_{S_{OOD}}}$ represents an environment where all API names and parameter names are completely different from $\mathcal{P_{S_{in}}}$. This distinction is crucial for our experimental design:
> * $\mathcal{P_{S_{in}}}$: Represents the initial API environment where our method learns to adapt to tool variability.
> * $\mathcal{P_{S_{OOD}}}$: Represents a completely new API environment, distinct from $\mathcal{P_{S_{in}}}$, used to comprehensively evaluate our method's adaptability, which serves as out-of-distribution tests (these APIs are new and agnostic for LLM).
>
> > Q2
>
> We appreciate your attention to detail regarding the results in Tables 2-4.
> * For most tasks, there are 100 test cases, except for "Coffee hard" which has 130 cases.
> This follows the setup of [1].
> And also following [1], the ".0" was retained for consistency across all tasks, aligning with the "Coffee hard" task's decimal representation. We will remove this if you think it's unnecessary. Thank you for this suggestion.
>
> * Regarding significant margins, we conducted three runs for each non-proprietary model and averaged the results in Table 2-4.
>
> [1] Zhuang, Y, et al. (2023). Toolqa: A dataset for llm question answering with external tools. Advances in Neural Information Processing Systems.
>
> > Q3
>
> The tool-update module is reset for each test instance. This design choice was made for several important reasons:
> * Resetting for each test case ensures that the order of test samples does not significantly impact performance.
> * Different test cases often require different tools. Accumulating updated API usage for all encountered tools in previous test instances could lead to unnecessarily long input sequences. Our goal is to evaluate the model's ability to independently handle potential tool variability, rather than relying on a growing memory of previously updated tools. This approach better simulates real-world scenarios where an agent might encounter tool variability at any time and needs to adapt on the fly.
>
> > Q4
>
> Thank you for your insightful question, which is indeed a crucial aspect of our work.
>
> In fact, $\mathcal{P_{S_{OOD}}}$ is specifically designed to simulate the real-world OOD scenario you've described. It features entirely different API names and parameter names compared to the training set $\mathcal{P_{S_{\text{in}}}}$, effectively presenting the model with completely new APIs. To our knowledge, our work is the first to explicitly consider tool variability. While we believe $\mathcal{P_{S_{OOD}}}$ offers a robust OOD test, we fully encourage the community to develop additional benchmarks that further challenge and evaluate tool learning models in dynamic environments.
>
> > Minor comments 1
>
> Thank you for your keen observation regarding Figure 2.
> Our method does not show a pronounced performance drop compared to the other methods, which is a key strength of our method. To this reason, we do not add an indicator here.
> However, we acknowledge that the absence of any indicator might be misleading. Based on your feedback, we have revised Figure 2 to more accurately represent the slight performance change in our method.
>
> > Minor comments 2
>
> You are correct in your assumption. This convention follows previous work in the field. We apologize for not making this explicit in the paper and appreciate your attention to detail. We have added a clear explanation of this formatting `in the title of Table 2` to ensure all readers can easily interpret the results.
>
> > Minor comments 3
>
> We sincerely apologize for this oversight. SFT stands for "Supervised Fine-Tuning."
> We will ensure that the full term is introduced at its first occurrence in the revised manuscript (`Line 81 in the title of Figure 2`). We appreciate your careful reading.
>
> We hope the above clarifications will help you better understand our work. We are happy to communicate with you further if you have any questions to enhance our manuscript.
>
> Sincerely,
> Authors of 2051

---

> ### Author Response · Authors · 2024-11-20
>
> Dear Reviewer VQDs,
>
> We hope this message finds you well. As the discussion period is drawing to a close, we wanted to reach out regarding our manuscript. We have submitted comprehensive responses to your insightful reviews and would greatly value any additional feedback you might have on our rebuttal.
>
> If there are any points that require further clarification or discussion, please don't hesitate to let us know. We remain committed to improving our work based on your expert guidance.
>
> Thank you for your dedication to the review process.
>
> Best regards,
>
> Authors of 2051

---

> > ### Comment · Area_Chair_XmvY · 2024-11-24
> > **Please respond to rebuttal ASAP**
> >
> > Dear reviewer,
> > The process only works if we engage in discussion. Can you please respond to the rebuttal provided by the authors ASAP?

---

> ### Comment · Reviewer_VQDs · 2024-11-25
> **Official Comment by Reviewer VQDs**
>
> Thank you for responding to my comments,
>
> I’ve read the updated manuscript and the reviews and corresponding comments to the other reviewers.
> Overall, I found the new manuscript easier to understand now and thank the authors for the changes. However, I suggest that the authors make it clear in the introduction (specifically line 77 ff. and line 90 ff.) that MCTS is only used to collect data for finetuning. Currently this is not clear until later in the paper. This could also be made clear in figure 1. Additionally, I found the new text added in line 269 ff. quite cumbersome to read and hard to understand. I would encourage the authors to improve those lines.
>
> Regarding **Q1**: I understand that the API changes $P_{S_{ood}}$ are different from the changes done in $P_{S_{in}}$, however, I am not convinced that the model really learns to adapt to changing APIs out-of-distribution but rather learns patterns to just replace old variables with new ones obtained from the feedback at inference time. It would be interesting to see how ToolEVO works on an API that is in $P_C$ but not in $P_{S_{in}}$ but then tested in $P_{S_{ood}}$. Nevertheless, I can still see the use in such abilities.
>
> A bigger question that I have after reading all the reviews is as follows: Your ablation study makes it clear that the self-reflection module plays and integral part for the performance of your method (even in the ablation study you still invoke the module for deprecation errors). I am not sure from the text nor from the reviews if the other models have access to this self-reflection process (you only mentioned that the other methods can use the update module). If the other models do have access to self-reflection, please make it clear in the text. Otherwise, my question is how the performance baselines would compare to ToolEVO with the self-reflection model.

---

> > ### Author Response · Authors · 2024-11-25
> >
> > Dear Reviewer VQDs,
> >
> > > Q1
> >
> > Thank you for your insightful suggestions. We have revised the manuscript according to your recommendations. Specifically, we **have clarified in Figure 1 and on line 81 that MCTS is solely used for finetuning**. We believe these changes will make this aspect of our methodology clearer from the outset. Additionally, we have **improved the text around line 269** to better articulate the distinct purporse of APIs in both the prompt and server settings, thereby enhancing the overall readability and comprehension of our objectives. We sincerely appreciate your valuable feedback and hope these revisions meet your expectations. If you have any further questions or suggestions, please do not hesitate to reach out. We are committed to producing manuscripts of the highest quality.
> >
> > > Q2
> >
> > Thank you for your thoughtful feedback. We agree that further benchmarking would enhance the evaluation of our model's performance, particularly in the scenarios you highlighted. Nonetheless, through our current work, we hope to raise awareness of the importance of dynamic environments for model adaptation. Your feedback is invaluable, and we are grateful for your contribution to improving our research.
> >
> > > Q3
> >
> > Indeed, all our baselines have access to both the self-reflection and tool update modules. However, **various models still encounter challenges for different reasons, such as the tendency to trust the prompt-provided tools**, as mentioned `in our error analysis (Appendix C.3)`. This underscores the importance of incorporating dynamic environments into the training trajectory. We appreciate your feedback and are committed to further exploring these dynamics to enhance model robustness.
> >
> > Thank you once again for your constructive feedback. We look forward to continuing the dialogue with you to further enhance our manuscript.
> >
> > Best regards,
> >
> > Authors of 2051

---

> > > ### Comment · Reviewer_VQDs · 2024-11-26
> > > **Official Comment by Reviewer VQDs**
> > >
> > > Thank you for further clarifications and adjusting the manuscript.
> > >
> > > The paper is now easier to read and while I believe the contribution is a small but good step towards tool variablity for LLMs worthy of publication.Therefore, I increase my presentation score from 2 to 3 and overall score from 5 to 6.

---

> > > > ### Author Response · Authors · 2024-11-26
> > > >
> > > > Dear Reviewer VQDs,
> > > >
> > > > Thank you very much for your positive feedback. We are pleased to hear that you find our contribution to be a valuable step towards addressing tool variability for LLMs. We hope to raise awareness of the importance of dynamic environments for tool learning, rather than focusing solely on static environments.
> > > >
> > > > We are committed to further enhancing the quality of our work and are open to any additional suggestions you might have that could help us improve the paper even further. Please feel free to share any other thoughts or recommendations.
> > > >
> > > > Thank you once again for your constructive feedback and support.
> > > >
> > > > Best regards,
> > > >
> > > > Authors of 2051

---

### Official Review · Reviewer_zgUw · 2024-11-04

**Soundness:** 2
**Presentation:** 2
**Contribution:** 2
**Rating:** 3
**Confidence:** 3

**Summary:**

The paper proposes ToolEVO, an algorithm for improving LLM tool use in dynamic environments. Specifically, the paper explores whether LLMs can call APIs to complete tasks, when the APIs can change over time. The change causes the API information in the static prompt to become outdated. The proposed method assumes the static prompt is not changeable and then proceeds to develop techniques to adapt the prompt and / or model using recently observed data. This is done via a combination of techniques (MCTS, Tool update, Self-reflection). The primary contributions of the paper are algorithmic and empirical. The paper also proposes a new dataset, ToolQA-D, constructed by random mutations of the ToolQA dataset. Experiments on ToolQA-D comparing ToolEVO using open-access LLMs to SOTA LLM baselines (closed and open access) are included along with ablation studies and other empirical analyses.

**Strengths:**

- The paper tackles an important and interesting problem of adapting LLMs to be able to invoke changing APIs correctly. This is a practical and important use case for LLMs. Progress here is likely to be of interest to the community.

- The paper considers a number of strong LLM baselines consisting of SOTA closed and open-access LLMs and performs a large set of computationally intensive experiments. While it's not clear if ToolEVO outperforms these, it's still useful to see relative performance. The results seem to suggest that LLM tool use is rapidly improving.

**Weaknesses:**

- Overall, I found the problem setup and proposed methods a little challenging to understand in detail. The implementation details of MCTS and its various enhancements (cached rollout, inference, computational costs) are not clearly described. More on this below.

- I didn't find the problem setup convincing. Restricting the LLM to only use P_C in the static prompt (Line 296) seems very limiting. Why can't the LLM's prompt be continuously optimized using recent API data in a separate process? That is what the ToolUpdate / SystemTool appears to do anyway. Please discuss this in more detail. Also, is it possible to include a baseline consisting of a proprietary LLM that is allowed to update its static prompt using the latest API invocation data, before each task?

- Given the above, the use of a full anytime planner like MCTS with cached rollouts is an interesting choice. As mentioned above, I'm not sure exactly how MCTS is used at inference time. Given that the state $s_t$ includes a full history, it's unclear to me if any tree node is visited more than once. Please consider including an illustrative example on a single task starting from the root. I wasn't able to find any discussion of computational costs (time, token), hyper-parameters and other standard MCTS implementation details. As a result, I found it very challenging to assess the algorithmic novelty and contributions of the proposed method.

- The methodology used to construct ToolQA-D from ToolQA isn't well motivated. Line 274 states "we employ GPT-4 to randomly modify the collected API usage". How exactly is this done? Why is this a good idea versus other mechanisms (e.g., using actual API versions of existing libraries)? Appendix A.3 doesn't say. Please motivate this choice and provide implementation details, assuming the goal is to propose ToolQA-D as a benchmark dataset for designing and evaluating tool-using LLMs.

- The presentation of the results in Table 2, 3 and 4 is a bit confusing. The bold formatting typically suggests best performance but that doesn't seem to be the case here. For example, on Line 335, ToolEVO's 30.3 score is in bold text but the higher score of 45.3 for Claude-3.5-Sonnet on Line 331 is not. Why is this? More generally, the proprietary LLMs, using only P_C, seem to perform best wrt Average-Hard (rightmost column) in Tables 2, 3 and 4. Is this correct? If yes, does that make the baseline proprietary LLM the best method? Please consider including a detailed discussion of the main results in Tables 2, 3 and 4 with an appropriate caption.

- The presentation and text gets a bit hand-wavy at times. Some examples below.
  - (Line 140) "rather than merely executing rigid tool invocations"
    - What does this mean? How do we know this is what the baseline LLM is doing?
  - (Line 398) "the stereotypes induced by Static SFT"
    - What stereotype is being referred to here?
  - (Line 406) "the model tends to lazily focus on how to use APIs provided in the prompt"
    - What does this mean?

- Overall, the paper seems to have a few major technical issues. As a result, I don't think it's quite ready for publication at this time.

**Questions:**

1. Using the Average-Hard column in Tables 2, 3 and 4, is it reasonable to claim that Claude-3.5-Sonnet outperforms all methods, including ToolEVO? If yes, why is the ToolEVO result highlighted using the bold formatting in Lines 335, 338, 351, etc.?

2. Assuming the prompts of the proprietary baselines in Tables 2, 3 and 4 are only shown the outdated API names and descriptions (P_C), is it possible to create a baseline that does a quick update of its prompt using the latest API names and descriptions before test-time evaluation / inference? How might this baseline perform?

3. How exactly is MCTS used at inference time? Please consider using an illustrative example task (API invocation).

4. What is the sample complexity and computational considerations (time and token costs) of ToolEVO? How does it compare with those of the proprietary LLMs?

---

> ### Author Response · Authors · 2024-11-15
>
> Dear Reviewer zgUw,
>
> We extend our sincere gratitude for your thorough evaluation and for recognizing the significance and interest of the problem we have addressed. Below, we offer detailed responses to each of your comments in hopes of alleviating any concerns:
>
> > W1
>
> We express our gratitude for your insightful feedback. The problem setup and our methods are summarized as follows:
> * In real-world scenarios, the update frequency of APIs (on the server) far exceeds that of the collection of API documentation (in the prompt). This discrepancy can result in mismatches between the prompt-provided APIs and server-deployed APIs, leading to a series of invocation errors. These errors are not related to the models' tool-invocation capabilities but rather to their adaptability to dynamic environments. We term this issue "tool variability."
>     - As discussed in `Lines 62-65`, a potential solution is the real-time collection of the latest APIs. However, this approach may introduce challenges, such as high latency in gathering API documentation and summarizing the latest API usage. This method also assumes the absolute correctness of API usage provided in prompts. We propose that models should adapt to dynamic environments akin to humans, rather than striving for absolute correctness of information within the prompts. This adaptability is the focal point of our work.
> * To evaluate the impact of tool variability, we introduce the first dataset specifically designed for tool dynamism, ToolQA-D. This dataset facilitates the exploration of performance variations across different methods and models in the presence of tool variability. Our findings indicate that tool dynamism significantly degrades the performance of existing methods.
> * In addressing this problem, we aspire to incorporate dynamic environments into the training episodes by leveraging MCTS, self-reflection, and tool update modules. This integration aims to enable models to recognize the potential for API changes and to develop the ability to invoke new APIs based on environmental feedback. Our experiments further validate the efficacy of our proposed approach.
>
> Regarding the implementation details, we have also included a detailed explanation and examples of the Cached rollout in `Appendix A.3` to further clarify our approach. Additionally, to assist in a deeper understanding of our methodology, we have provided pseudocode for our customized MCTS in Appendix A.4 of the revised manuscript. We hope these additions will offer readers a clearer and more comprehensive view of our proposed solution, and we are open to any further suggestions you may have.
>
> > W2
>
> Thank you for your insightful comments. We'd like to clarify a crucial misunderstanding:
>
> * While attempting to access the latest and correct API invocation before each task is a potential approach for addressing tool variability, `as discussed in Lines 62-65`, it may introduce significant challenges, such as latency issues, especially when managing numerous APIs. It is important to note that ToolUpdate fundamentally differs from the approaches mentioned. `As discussed in Lines 241-245`, ToolUpdate is a memory mechanism enabling the model to summarize and modify the API usage in the prompt based on newly acquired knowledge. We
>
> * Furthermore, it's important to emphasize that all baselines in our study, including proprietary LLMs, were indeed integrated with ToolUpdate/SystemTool, `as mentioned in Lines 920-922 and Appendix D.2`. However, our error analysis in `Appendix C.3` reveals that these models still struggle to effectively leverage new tools based on environmental feedback, encountering issues such as invalid invocation, repeated use of deprecated tools, and ignoring environmental feedback. Thus, it is crucial for LLMs to recognize tool variability (a central focus of our paper), which has often been overlooked in previous work.
>
> > W3
>
> Thank you for your valuable comments. We'd like to clarify some misunderstandings:
> * As noted in `Lines 316-317`, during the inference phase, all methods presented in our paper use **greedy decoding** for a fair comparison, rather than MCTS. We believe that using MCTS for inference presents significant challenges in tool variability:
>     1. MCTS introduces considerable latency during inference.
>     2. Effective MCTS inference requires a reliable value model. However, in scenarios of tool variability, the value model is susceptible to reward hacking, leading to decreased performance, as discussed in `Lines 830-835`.
> * To address your concerns, we have included pseudocode for our customized MCTS in Appendix A.5. Our primary objective is to leverage MCTS to balance exploitation and exploration, allowing for autonomous data collection. Using the instructed data collected through MCTS, ToolEVO is designed to enable the model to recognize outdated tools and invoke new ones based on environmental feedback, rather than employing MCTS during inference.

---

> ### Author Response · Authors · 2024-11-15
>
> > W4
>
> We appreciate your valuable suggestion. We have added the specific prompt for modifying API usage via GPT-4 as Appendix D.1 in our revised version to address your concerns.
>
> In `Section 4 of our original paper` (or Appendix B.1 in our revised paper), we outline four reasons for constructing the first benchmark of tool variability based on ToolQA rather than other datasets. Regarding your concerns about "using actual API versions of existing libraries", it's important to note that existing tool learning benchmarks lack detailed records of different API versions. Tool variability have been consistently overlooked in the existing literature, which is a key aspect that our work seeks to emphasize.
>
> > W5
>
> We apologize for any confusion caused. We will clarify this in the title of Table 2 in the revised version, where bold text indicates the best performance and an underline denotes the second-best performance among open-source models. Thank you for highlighting this.
>
> Given that the proprietary models have significantly more parameters than other methods (for instance, our model has only 7B parameters), we present them solely as a performance reference. Moreover, the $\mathcal{P}_c$ environment indicates that the prompt is consistent with the API used on the server, without any tool variability, which is a setup common in prior works. Hence, it is unsurprising that Claude-3.5-Sonnet performs well in this context.
>
> We included this setting to comprehensively assess our ToolEVO's effectiveness.
> Notably, our method has not been fine-tuned on the APIs in the $\mathcal{P}_c$ environment (as shown in Table 1). The results in Table 2 demonstrate that focusing on tool variability does not adversely impact performance in static environments ($\mathcal{P}_c$). We hope this explanation resolves your concerns.
>
> > W6
>
> * For Q1 in W6:
>
> Existing methods primarily focus on executing tool invocation, often neglecting tool variability. This oversight can lead to a decline in model performance within dynamic environments, as evidenced by our findings presented in Tables 2-4.
>
> * For Q2 in W6:
>
> As stated in `Line 399`, due to space constraints, we elaborate on this phenomenon in our error analysis (Appendix C.3). The term "stereotype" refers to the tendency to concentrate solely on current tool invocation patterns, neglecting the adaptation required for dynamic environments. This stereotype can lead to an overconfidence in the tool usage specified in the prompt. Consequently, when the model encounters errors as indicated by environmental feedback, it may persist in using the tools from the prompt without reassessing whether these tools are outdated.
>
> * For Q3 in W6:
>
> This is one of the conclusions drawn from our experiments: when tool dynamics are absent during the training phase, the model tends to adopt a "lazy" attitude, focusing exclusively on the tool usage specified in the prompt. This explains the noticeable decline in performance of existing methods when faced with tool variability.
>
> We believe the clarifications provided above address your concerns.
>
> > W7
>
> Thank you for your valuable feedback. We have provided further clarifications on all the confusions and misunderstandings you mentioned, aiming to address your concerns.
>
>
> > Q1
>
> As discussed in W5, we will clarify this in the title of Table 2 in the revised paper. Thank you for your valuable suggestion.
>
> > Q2
>
> As described in Section 5.2, Table 2 was specifically created to align with the setup you mentioned. The "using the latest API" you referred to implies that the APIs in the prompt are consistent with those deployed on the server, which aligns with the setup of Table 2. We hope this clarifies your concerns.
>
> > Q3
>
> As noted in `Line 316`, we do not use MCTS during inference. Greedy decoding is employed for all methods in our paper to ensure fair comparison.
>
> > Q4
>
> With greedy decoding used for all methods, computational time and costs are quite similar. However, proprietary LLMs can be affected by Queries per Second (QPS) or network latency. Therefore, we compare our approach with Static-SFT as an exmaple:
>
> | Method    | Time Per Question | Avg. Steps |
> |-----------|-------------------|------------|
> | Static-SFT| 7.8s              | 9.6        |
> | ToolEVO   | 6.6s              | 7.4        |
>
> It's noteworthy that ToolEVO demonstrates superior inference speed. In contrast, Static-SFT generates more incorrect steps due to its failure to adapt to tool variability, resulting in a greater number of inference steps.
>
> We believe our explanations have clarified your concerns and provide a clearer understanding of our work. Thank you for your constructive feedback. We welcome further discussions to enhance and refine our work.
>
> Sincerely,
>
> Authors of 2051

---

> ### Author Response · Authors · 2024-11-20
>
> Dear Reviewer zgUw,
>
> Thank you for your valuable feedback on our submission. We have provided detailed responses to your comments and addressed your concerns. As this is the final week of the discussion period, we kindly request your review of our responses and welcome any further questions or clarifications you might have. Your input is crucial for improving our work, and we greatly appreciate your time and effort.
>
> Looking forward to your feedback!
>
> Best regards,
>
> Authors of 2051

---

> > ### Comment · Area_Chair_XmvY · 2024-11-24
> > **Please respond to the rebuttal ASAP**
> >
> > Dear reviewer,
> > The process only works if we engage in discussion. Can you please respond to the rebuttal provided by the authors ASAP?

---

> ### Comment · Reviewer_zgUw · 2024-11-25
> **Response to rebuttal**
>
> - I thank the authors for their detailed responses. A number of my questions remain unaddressed. Details below.
>
>     - The setup remains unconvincing to me. The sentence "in all experiments, the LLM can only access P_C in the prompt." does not seem like a reasonable setup to me. I'd really like to better understand why LLM parameter updates via SFT is a more reasonable approach than an updated prompt (P_C + P_S) based on recent traces / logs.
>
>     - I think a comparison is warranted to test the claims in Lines 62-65 justifying the setup. Is there a way to evaluate this empirically?
>
>     - How would a baseline allowed to use the last K trajectories to construct a new P_C before evaluation perform?
>
>     - What are the computational tradeoffs between the above baseline and the proposed method wrt token and time costs?
>
>     - Thanks for the MCTS usage and inference clarifications. Since MCTS is only being used offline, what might be other ways to collect trajectory data offline and how would they perform? As an example, what is the data quality of API trajectory sequences generated by standard software testing strategies (e.g., random testing, regression testing)?
>
>     - Might it be possible to update P_C by executing a short test suite just before evaluation? How might this perform?
>
>     - The response to the question about experimental performance versus closed-source LLMs (Claude-3.5-Sonnet) is unconvincing. The results of the closed-source LLMs are not just better on the "Hard" instances in the P_C server setting in Table 1 but seem to be better in the other two settings (Table 2 and Table 3) as well. Is my understanding correct?
>
>   - I have a few additional questions and concerns but these are the major ones so will stop here for now.

---

> > ### Author Response · Authors · 2024-11-25
> >
> > Dear Reviewer zgUw,
> >
> > Thank you for your continued engagement and for highlighting areas that require further clarification.
> >
> > > Q1
> >
> > Firstly, there might be a misunderstanding regarding our approach.
> > The parameter updates in our ToolEVO is **not** for mastering latest API usage, but are primarily focused on **adapting to dynamic environments and tool variability**. We aim to enabling the model to be aware of a dynamic environment and tool variability.
> >
> > Secondly, our approach does not exclude the potential benefits of prompt updating; rather, we view these methods as complementary. Here’s how they differ and can work together:
> > 1. Prompt Updating: As you noted, updating the prompt based on recent traces or logs is crucial to maintain the stability and accuracy of API demonstrations. However, this approach can result in a scenario where users encounter errors before the prompt is updated, resulting in failures due to outdated tool information. **This is a critical issue our work seeks to address.**
> > 2. **Dynamic Environment Awareness**: Our method emphasizes the importance of the model's ability to adapt to dynamic environments. We believe that enabling the model to autonomously adjust its tool usage based on environmental feedback (such as invoking errors, tool deprecation, and basic information about new tools) is a more promising approach. This mimics human adaptability and allows for a more robust handling of tool variability. While this doesn't negate the benefits of prompt updates, it underscores the significance of adaptive learning in dynamic tool environments.
> >
> > Our intention with this work is to **highlight the importance of adaptability in tool learning, which can be combined with prompt updates**.
> >
> > Thank you again for your valuable feedback. We look forward to further discussions and are eager to address any additional questions you might have.
> >
> > > Q2
> >
> > Honestly, conducting a comprehensive empirical evaluation of Lines 62-65 is challenging during the rebuttal period.
> > In our future work, we plan to develop a framework that allows the model to retrieve the latest documentation before tool invocation. This framework would compare the current prompt-provided APIs with latest tool documentation and synthesize the differences to adapt the tool usage accordingly. However, constructing such a benchmark and designing a pipeline for this evaluation is a significant undertaking beyond the scope of the rebuttal period. We apologize for not being able to present this empirical analysis at this time.
> >
> > However, we **have provided the upper bound of this evaluation** in our paper. The approach described in Lines 62-65 of our paper essentially aligns with executing a brief test suite to update P_C prior to evaluation, as you mentioned in your question 6. **The best outcome of this approach** is when the promprt-provided APIs perfectly match with server-deployed APIs, which is demonstrated in our experimental results in Table 2.
> >
> > On the other hand, we believe this approach of lines 62-65 complements the perspective of our method, rather than contradicts it. Just as humans adeptly adjusting to dynamic environments, we aspire for models to develop similar adaptability. This reduces the need for manually maintaining a static environment for the model in the prompt, which is a key message of our work, but it does not mean that we deny that prompt updates are not feasible. **We believe they complement each other**.
> >
> > > Q3
> >
> > Thank you for your question regarding the use of recent trajectories to construct a new $\mathcal{P}_{C}$ before evaluation. I would like to reiterate that updating the prompt is indeed a feasible approach. The scenario you described, where "the last K trajectories are used to update P_C", represents an ideal state where the prompt-provided APIs aligns perfectly with the current server-deployed APIs. This would eliminate any issues related to outdated tools.
> > **Our results in Table 2 aim to demonstrate precisely this scenario**. The experiments outlined in Table 2 reflect the outcome where prompt updates ensure consistency between the prompt-provided APIs and the server-deployed APIs. Thus, **it effectively addresses the scenario you are interested in testing**.
> >
> > I hope this explanation clarifies the setting of our experiments. We are eager to engage in further discussions to address any remaining questions or concerns you might have.

---

> ### Author Response · Authors · 2024-11-25
>
> > Q4
>
> To address your concerns regarding the computational trade-offs, I have provided a comparison of token counts and time costs between our method and the baselines:
>
> | Model                    | Avg Tokens | Avg Time (seconds) |
> |--------------------------|------------|--------------------|
> | Claude-3.5-Sonnet-20240620 | 971.84     | N/A                |
> | GPT-4-Turbo-2024-04-09   | 1334.0     | N/A                |
> | GPT-4o-Mini              | 3266.4     | N/A                |
> | Static-SFT               | 1091.47    | 7.8s               |
> | Ours                     | 1031.21    | 6.6s               |
>
> I hope this comparison clarifies the computational aspects of our approach.
>
> > Q5
>
> Thank you for your question about alternative methods for collecting trajectory data offline. One possible approach is to use reject sampling. However, a significant limitation of reject sampling is **its inability to allow the model to improve its decision-making through feedback**. This limitation is precisely why AlphaGo employs MCTS for collecting game trajectories; MCTS enables iterative learning and decision refinement, which reject sampling does not support. I believe you can appreciate why AlphaGo favors MCTS over reject sampling for these reasons.
>
> Regarding the standard software testing strategies you mentioned, such as random testing and regression testing, it is unclear how effectively these methods can facilitate an agent in collecting API data. These strategies are traditionally designed to evaluate the robustness of software, rather than to train agents on dynamic environment. As such, they address a different problem and may not be directly applicable to our context.
>
> Thank you for your continued feedback. We look forward to further discussions to explore these ideas in more depth.
>
> > Q6
>
> As I mentioned in my response to question 3, **Table 2 of our paper directly addresses the scenario you described**.Our experiments have already taken this aspect into account.
>
> Furthermore, as we discussed earlier, **Table 2 represents the theoretical upper bound of performance**, where the prompt-provided APIs and server-deployed APIs are perfectly aligned. **This is the upper bound for the methods you are interested in (as mentioned in your lateset questions 1, 3, and 6)**. Achieving such a theoretical upper bound requires us to **rigorously ensure that the APIs in the prompt and on the server remain completely consistent (this is a static environment**, we understand that you prefer to have the API usage in the prompt updated in real-time, but for the model, this constitutes a static environment where the APIs provided in the prompt are always correct. **The model does not perceive the dynamic environment in this setting**).
>
> While we acknowledge that this is a feasible approach, our method focuses on **enabling the model to adapt directly to tool variability in dynamic environments**, much like humans do. Since the approach you mentioned **requires the prompt to be correct at all times**, we find it to be too stringent and not reflective of real-world applications. Therefore, we have highlighted **the often-overlooked issue of dynamic environments in tool learning**, or tool variablity. In our setup, as shown in Tables 3 and 4, our ToolEVO **closely** approaches the theoretical performance upper bound in Table 2, demonstrating the effectiveness of our approach.
>
> Thank you for your attention, and I am happy to discuss further if you have additional questions or need more clarification.
>
> > Q7
>
> You are correct in observing that Claude-3.5-Sonnet performs better in almost experinment. However, **it's important to note that our model is a 7B model, smaller than Claude-3.5-Sonnet**. As we discussed in your inital question (Weakness 5), the experiments involving Claude-3.5-Sonnet were included primarily as a performance reference. **We believe it is challenging for a 7B model to surpass the performance of Claude-3.5-Sonnet**. Hence, it is unsurprising that Claude-3.5-Sonnet performs well in this context. It is noteworthy that **our model achieves performance that is closest to Claude-3.5-Sonnet among other baselines**.

---

> > ### Author Response · Authors · 2024-11-25
> >
> > > Q8
> >
> > Thank you very much for engaging in the discussion of our work and for contributing valuable questions that have helped improve our research. Beyond addressing your specific inquiries, we would like to take this opportunity to engage in a deeper conversation with you.
> >
> > It seems there might be a misunderstanding regarding the focus of our work. Your comments suggest a perspective that centers on maintaining up-to-date tool demonstrations in prompt for the model to use, which is indeed a valid approach as we discussed in Lines 62-65. Our paper does not dismiss this method; rather, we aim to highlight an complementary perspective: enabling the model to adapt to dynamic environments on its own, like humans do. This approach mirrors human-like adaptability and decision-making in dynamic environment.
> >
> > We believe that **current tool-learning methodologies often emphasize a model's mastery over existing tools while overlooking the dynamic nature of the environment**. We argue this is a risky oversight. Models should not be confined to a **"protective"** setting where they rely solely on static updates. For instance, your suggestions to "use the last K trajectories to construct a new P_C before evaluation" or "update P_C by executing a short test suite just before evaluation" focus on ensuring the prompt's tool usage remains stable and accurate, **maintaining a controlled environment**.
> >
> > Our core message is to **raise awareness of the importance of dynamic environments in model training** and to **encourage a shift towards developing models capable of adapting to dynamic environment autonomously**. We hope this clarifies our intention and the significance of our work.
> >
> > Thank you again for your thoughtful engagement. We look forward to continuing this dialogue and exploring these ideas further with you.
> >
> > Best regards,
> >
> > Authors of 2051

---

> ### Author Response · Authors · 2024-11-28
> **Follow-up on Rebuttal Discussion and Further Feedback**
>
> Dear Reviewer zgUw,
>
> I hope this message finds you well. We appreciate the time and effort you have put into reviewing our paper and providing valuable feedback. As the rebuttal deadline is fast approaching, we would like to kindly follow up and see if there are any remaining points or concerns that we could address.
>
> We are eager to engage in further discussion and would greatly appreciate any additional thoughts or suggestions you might have. Please do not hesitate to share any further feedback, and we will ensure a prompt response.
>
> Thank you once again for your time and consideration.
>
> Best regards,
>
> Authors of Submission 2051

---

> ### Author Response · Authors · 2024-12-02
>
> Dear Reviewer zgUw,
>
> We would like to express our sincere appreciation for the time and effort you have invested in reviewing our paper. Your feedback has been extremely helpful and insightful.
>
> As the rebuttal deadline approaches, we would like to kindly check in regarding any remaining concerns or points you might still have. We are eager to continue the discussion and address any issues in order to further improve our work.
>
> If you have any additional thoughts or suggestions, we would greatly appreciate hearing from you. We are committed to providing timely responses and value your guidance throughout this process.
>
> Thank you once again for your time and support.
>
> Sincerely,
>
> Authors of 2051

---

### Official Review · Reviewer_Haos · 2024-11-08

**Soundness:** 2
**Presentation:** 2
**Contribution:** 2
**Rating:** 6
**Confidence:** 3

**Summary:**

Large language models (LLMs) are typically trained on stationary datasets, but the world is constantly evolving. For example, the API of a programming tool can be updated or deprecated in new releases. An LLM trained on the old version of the API will produce incorrect outputs when interacting with the new API. This paper proposes a framework for adapting large language models to a variable external environment, in particular a variable API / tool.

For an LLM to adapt to new APIs different from those in the training set, it needs to interact with the environment and receive feedback. To this end, the authors propose to use online Monte-Carlo Tree Search (MCTS) to generate language actions to execute in the environment. The environment then returns the API outputs/error messages as well as a 0/1 reward at the end of the interaction sequence as feedback. In addition to MCTS, the authors propose two more mechanisms to improve adaptation: (1) self-reflection, where the model tries to explain the reason for encountering an error (as opposed to stopping right there); (2) tool-update, where the model generates a description of the updated API and add it to the context.

The authors evaluate their method, ToolEvo, on a curated benchmark of API / tool variability. There are three sets of APIs, $\mathcal P_{c}$, $\mathcal P_{s_\text{in}}$  and $\mathcal P_{s_\text{OOD}}$. $\mathcal P_{c}$ is the API seen in the training set, $\mathcal P_{s_\text{in}}$ is a slightly modified API that uses terminology seen in $\mathcal P_{c}$, and $\mathcal P_{s_\text{OOD}}$ is completely out of domain. Compared to a suite of baselines that are either only pretrained or supervised finetuned on $\mathcal P_{c}$, ToolEvo demonstrates significantly better adaptation capability, achieving higher success rates across the benchmark.

To summarize, the paper makes two contributions: (1) it proposes a framework for adapting LLMs to variable external environment, (2) it introduces a benchmark to evaluate such adaptation capabilities.

**Strengths:**

1. This paper addresses an important research problem in the community, i.e., adapting LLMs to an evolving external environment.
2. The proposed approach based on MCTS is grounded in a rich literature, and the self-reflection and tool-update mechanisms are novel contributions.
3. ToolEvo exhibits strong empirical results on the benchmark, outperforming a suite of proprietary, open-source, and fine-tuned LLMs.
4. The released benchmark could be useful for future research in this direction.

**Weaknesses:**

1. The setting is rather contrived. The authors curate three predefined sets of APIs, but the APIs don't change *within an evaluation episode*. So performing well on the test set doesn't necessarily mean the method adapts to a *constantly* evolving environment. In fact, The proposed approach wouldn't apply to a constantly evolving environment, as the Q value in the MCTS is not adaptive.
2. The evaluation is potentially unfair. For a LLM to adapt to a new set of APIs, it needs to receive feedback. The baselines only have access to the API outputs and error messages, but ToolEval also has access to a reward at the end of each episode, which is privileged information. The authors don't compare to methods that make use of the reward information.
3. The main contribution might not be the reason for the performance gain. According to Table 5, once they remove self-reflection and tool-update, the performance is about the same as supervised fine-tuning. This suggests MCTS may not contribute to the overall performance, despite it being introduced as a main contribution.
4. The presentation has much room for improvement. To a general reader, too much context is deferred to the appendix, making the initial read particularly rough.

**Questions:**

1. How much overhead does MCTS add to the inference time?
2. Can you compare ToolEvo to a method that uses online RL to adapt to the new API? If it's not possible, why?
3. Can you compare ToolEvo to a version of the Static-SFT combined with self-improve and tool-update? The goal is to understand if MCTS contributes to the overall performance. According to Table 5, the main performance gain comes from self-improve and tool-update, and the MCTS-only version gets about the same performance as Static-SFT.
4. The overall algorithm is quite vague. I would suggest adding an algorithm box in the main text.
5. In line 275, I would suggest adding a description of $\mathcal P_{s_{\text{in}}}$  and $\mathcal P_{s_{\text{OOD}}}$ right away instead of referring to the appendix.
6. According to Section 5.1 Implementation Details, the models are fine-tuned on a dataset of interactions with $\mathcal P_{S_{in}}$. Can you expand on the reason behind this?

---

> ### Author Response · Authors · 2024-11-15
>
> Dear Reviewer Haos,
>
> We sincerely appreciate your detailed review and the recognition of the important research problem we address, along with our proposed method and valuable benchmark.
> Below, we address each of your comments in detail.
>
> > W1
>
> Thank you for your insightful comments. Our work emphasizes the discrepancy between APIs provided in prompts and those deployed on servers. Our approach aims to enable models to interact effectively within dynamic environments and adapt by invoking new APIs based on environmental feedback. The occurrence of API changes with an evaluation episode implies that the API modifications happen precisely when the model invokes with the API. We consider this to be highly coincidental. We believe that inconsistencies between prompt-provided APIs and server-deployed APIs are more likely to occur in these dynamic settings. Moreover, we've demonstrated the robustness of our method in out-of-distribution scenarios, such as $\mathcal{P_{s_\text{OOD}}}$.
> Additionally, it's important to note that MCTS is not used during the inference phase; it is solely employed for data collection during training. Thus, the concern regarding non-adaptive Q-values does not arise.
>
> > W2
>
> As mentioned in `line 316`, all methods, including our ToolEVO, utilize `greedy inference`, and therefore, ToolEVO does not leverage any additional reward information. As discussed in `line 920`, all methods have access to API outputs, error messages, and even the tool update module. Hence, all experiments are conducted under fair comparison conditions.
>
> > W3
>
> Our MCTS is not employed for inference. In our approach, MCTS balances exploration and exploitation, improving the interaction between the LLM and dynamic environments. It is combined with self-reflection and tool-update mechanisms to facilitate adaptation to dynamic environments and to collect trial-and-error experiences for training. Thus, removing self-reflection and tool-update results in a method akin to static SFT.
>
> > W4
>
> Thank you for your thoughtful suggestions. We acknowledge this area for improvement. Our work tackles an important yet often overlooked problem in tool learning: tool variability. We have introduced the first benchmark to evaluate models' performance in tool variability and also proposed a method to address this challenge. As you noted, the comprehensiveness of our work required us to position some content in the appendix.
>
> To better cater to general readers, we have made the following revisions:
> * We moved the content on dynamic environments from Appendix A.2 to Section 2.2, enhancing the introduction of background knowledge and context.
> * We relocated the motivation for constructing our initial benchmark of tool variability to Appendix B, and integrated the data statistics originally in Appendix A.3 into Appendix B as well.
> * The settings and objectives of the three API environments ($\mathcal{P_{c}}$, $\mathcal{P_{s_{\text{in}}}}$, $\mathcal{P_{s_{\text{OOD}}}}$) in our ToolQA-D dataset have been shifted to Section 4, aiding readers' understanding of the experimental setup.
> * Additional minor adjustments, such as clarifying the meaning of bold and underline format in Table 2, have also been made.
>
> We hope these changes will offer a more accessible reading experience for a general audience. Thank you once again for your valuable feedback. We are committed to refining our work for a higher quality presentation.

---

> ### Author Response · Authors · 2024-11-15
>
> > Q1
>
> As discussed in response to W2, we have **not** incorporated MCTS into the inference phase. In scenarios of tool variability, training a value model to support MCTS inference poses challenges due to environmental changes stemming from tool variability. Such changes can potentially lead the value model into reward hacking, complicating effective adaptation.
>
> > Q2
>
> To the best of our knowledge, there are currently no online RL methods tailored to adapt to tool variability. We are pioneering the consideration of tool variability in tool learning. However, we agree that online RL is a promising avenue, and we hope to explore this direction in future work. Thank you for your insightful comments.
>
> > Q3
>
> As discussed in response to W2, combining Static-SFT with self-reflection and tool-update modules essentially results in our ToolEVO approach, as supported by our ablation studies. We believe these discussions help clarify any misunderstandings you might have had.
>
> > Q4
>
> Thank you for your suggestion. Due to space constraints in the main text, we have included the algorithm pseudocode in `Appendix A.4`.
>
> > Q5
>
> Thank you for your suggestion. We have moved the relevant section to line 275 and made modifications to clarify the description. For more details, please refer to our response in `W4`.
>
> > Q6
>
> Our work emphasizes incorporating environmental feedback into the trial-and-error process through self-reflection, tool-update, and MCTS modules. Thus, we focus on actively interacting with dynamic environments, integrating tool variability and environmental feedback into the Chain of Thought (CoT) process. This approach enables the model to recognize tool changes and learn to manage them effectively. Previous work primarily emphasized learning to invoke given tools in static environments. Additionally, we introduce an out-of-distribution environment, $\mathcal{P_{s_{\text{OOD}}}}$, to fairly evaluate the algorithm's performance.
>
> Thank you once again for your insightful reviews. We hope these clarifications enhance your understanding of our work. Please feel free to reach out for further discussion, as we are eager to improve our manuscript.
>
> Sincerely,
>
> Authors of 2051

---

> ### Author Response · Authors · 2024-11-20
>
> Dear Reviewer Haos,
>
> We sincerely appreciate your thoughtful reviews of our manuscript. We have carefully addressed all the points raised in our rebuttal and made several improvements to our work based on your suggestions. As we are approaching the end of the discussion period, we would be grateful if you could take a moment to review our responses and share any additional thoughts or concerns you may have.
>
> Your expertise and insights are invaluable to us, and we are eager to engage in further discussion to enhance the quality of our work.
>
> Thank you for your time and consideration.
>
> Best regards,
>
> Authors of 2051

---

> > ### Comment · Area_Chair_XmvY · 2024-11-24
> > **Please respond to the rebuttal ASAP**
> >
> > Dear reviewer,
> > The process only works if we engage in discussion. Can you please respond to the rebuttal provided by the authors ASAP?

---

> > ### Comment · Reviewer_Haos · 2024-11-25
> >
> > Thanks for addressing my questions and incorporating some of the feedback into the revised paper.
> >
> > > As mentioned in line 316, all methods, including our ToolEVO, utilize greedy inference, and therefore, ToolEVO does not leverage any additional reward information.
> >
> > In line 130 of the revised manuscript, you state that the task completion state (a -1/1 reward) is given as feedback. So ToolEVO uses reward information during fine-tuning. Do other baselines leverage this reward information during fine-tuning?
> >
> > > Our MCTS is not employed for inference.
> >
> > Thanks for the clarification. I was under the impression that MCTS is employed during inference. Looking at Algorithm 1 in the revised manuscript, it is clear that MCTS is only used to collect data for fine-tuning. I think it would make the paper much clearer if you emphasized in Section 3 or earlier that the proposed method is used for finetuning, not online adaptation.
> >
> > That said, I'm confused by the return value of Algorithm 1. Algorithm 1 returns the expansion tree for this task description $\mathcal D$. How is this expansion tree used afterward? Is it used to supervised finetune the policy $\pi$?

---

> ### Author Response · Authors · 2024-11-25
>
> Dear Reviewer Haos,
>
> We sincerely thank you for your thoughtful questions, which have helped us improve the clarity of our manuscript.
>
> > Q1
>
> We appreciate your insightful observation. To ensure a fair comparison, we deliberately constructed a strong baseline, **Static-SFT**, which incorporates the same reward information during fine-tuning. As detailed in `lines 315-316`, Static-SFT also leverages reward information during fine-tuning and employs MCTS to collect data but operates in a static environment.
> Notably, ToolEVO's superior performance over this strong baseline substantiates our key contribution: the significance of dynamic environments in tool learning, rather than merely focusing on mastering existing tools.
>
> > Q2
>
> Thank you for your valuable suggestions. We have revised `Figure 1`, `line 82`, and  `Section 3.1 (Lines 156-157)` to explicitly emphasize that our method is primarily focused on fine-tuning rather than online adaptation.
>
> Regarding Algorithm 1, we appreciate the opportunity to clarify its implementation details. The expansion tree serves as a data collection mechanism where we can extract different trajectories for different training paradigm.
> In fact, **multiple training paradigm can be employed by this expanded Monte Calro tree**, including: (1) extracting positive trajectories (successful paths from root to leaf nodes) for supervised fine-tuning, (2) constructing positive-negative sample pairs for preference learning methods such as DPO, or (3) implementing RLHF by training reward models with positive-negative sample pairs. As our research primarily focuses on **highlighting the crucial role of dynamic environments in tool learning**, we adopt the most straightforward approach, supervised fine-tuning, to improve the model's capability to adapt to tool variability.
>
> To enhance clarity, we have `revised Line 877-883` in Algorithm 1 to better articulate the various potential applications of the Monte Carlo tree in the training process. This modification helps readers better understand how the expansion tree contributes to the overall learning framework while maintaining our emphasis on the significance of dynamic environments in tool learning scenarios.
>
> We hope these clarifications effectively address your questions and provide a clearer understanding of both the scope of our method and the specific role of the expansion tree in our training pipeline. We are thankful for your detailed review that has helped us improve the presentation of our work.
>
> We welcome any further discussions and would be delighted to provide additional clarification if needed. Your feedback is invaluable in helping us improve our work.
>
> Best regards,
>
> Authors of 2051

---

> > ### Comment · Reviewer_Haos · 2024-11-26
> >
> > Thanks for the clarifications. All of my questions have been addressed, and I will increase my score accordingly.

---

> > > ### Author Response · Authors · 2024-11-26
> > >
> > > Dear Reviewer Haos,
> > >
> > > Thank you very much for your thoughtful feedback and for acknowledging the clarifications we provided.
> > >
> > > We are committed to continuously improving our work and are open to any further suggestions you might have that could enhance the paper even more. If you have any additional thoughts or feedback, please feel free to share them with us.
> > >
> > > Thank you once again for your support and constructive comments.
> > >
> > > Best regards,
> > >
> > > Authors of 2051

---

### Official Review · Reviewer_hYZV · 2024-11-08

**Soundness:** 3
**Presentation:** 2
**Contribution:** 2
**Rating:** 1
**Confidence:** 5

**Summary:**

While LLMs have been equipped to interact with external environments, improving their functionality, in many cases, these "tools" or external environments evolve in an unpredictable way inducing incorrectness in the LLM output. This paper presents a framework called ToolEVO that uses MCTS (Monte Carlo Tree Search) to acquire feedback from LLMs through autonomous exploration and interaction. The feedback is then used to adapt the LLM to the changing environment.

**Strengths:**

1. Considers adaptation of LLMs to the changing environments which is a real problem for existing LLMs

**Weaknesses:**

1. The method is computationally inefficient and unrealistic in a real-world setting
2. A simple method of accessing the APi's through a proxy can do the same job in a much more efficient way.
3. There have been a lot of work on using design patterns to deal with evolving environments. The authors seem to be unaware of that literature. As a result, they have designed a cumbersome method that is not likely to work in practice.
4. In addition, the LLM itself might be modified. Thus the adaptation module needs to be separate from the LLM (in other words, the LLM does not need tom adapt). Tuning the adaptation module (which will be much smaller) should do the job.
5. The design is flawed.

**Questions:**

1. Think about a more efficient design.
2. The LLM should not be adapted. It is the adaptability module (which is the interface) that should be adapted.
3. Need precise runtime statistics (including response times).

---

> ### Author Response · Authors · 2024-11-15
>
> Dear Reviewer hYZV,
>
> We sincerely appreciate the time and effort you have dedicated to reviewing our work. We are grateful for your insights and would like to address each of your comments in detail to help alleviate any concerns you may have.
>
> > W1
>
> Thank you for your feedback. However, we would really appreciate your clarification regarding which specific aspects of our algorithm led to the assessment of it being computationally inefficient and unrealistic. It is important to note that our MCTS is not utilized during the inference phase; rather, it serves to enhance the LLM's interaction with dynamic environments, thereby enabling the model to develop the capability to recognize outdated tools and adapt to new tools based on environmental feedback. As outlined in `Line 316 of our manuscript`, our method employs **greedy decoding** during inference. Consequently, we are uncertain which components of our approach give rise to concerns about computational inefficiency and practical feasibility. We welcome further discussion to address these specific concerns and clarify any potential misunderstandings.
>
> > W2
>
> We sincerely appreciate your valuable feedback. The proxy approach heavily relies on the manual maintenance of proxy servers, which incurs higher costs in real-world scenarios. While ToolBench[1] implemented a tool server solution, the frequent API updates and subscription expirations in practice necessitate considerable human intervention for maintenance. This often renders the tool server non-functional, presenting a persistent challenge that remains unresolved in the current literature.
> Stable ToolBench[2] proposed a hybrid approach combining caching and GPT-4 simulation to create a virtual environment for evaluating tool learning. However, this solution is primarily designed for evaluation purposes and does not address the fundamental challenge of enabling models to interact with real APIs.
>
> The essence of the proxy is an emphasis on achieving a stable environment. In contrast, the primary distinction of our work compared to previous studies lies in enabling the model to recognize the dynamic nature of the environment and the evolving nature of tools. Our approach focuses on learning how to utilize tools effectively by receiving feedback from the environment, akin to human behavior. We believe our explaination clarifies your concerns.
>
> [1] Qin, et al. ToolLLM: Facilitating Large Language Models to Master 16000+ Real-world APIs. ICLR, 2024.
>
> [2] Guo et al. StableToolBench: Towards Stable Large-Scale Benchmarking on Tool Learning of Large Language Models. ACL Findings 2024.
>
> > W3
>
> Thank you very much for your valuable comments. To the best of our knowledge, there is currently no existing work that addresses dynamic environments and tool variablity in tool learning. If possible, we would greatly appreciate it if you could provide specific relevant literature to help us further enhance our work. Once again, we thank you for your insights.
>
> > W4
>
> Thank you for your feedback. Training a small number of additional parameters through techniques such as adapter tuning or LoRA, without modifying LLM itself, is indeed a viable approach. However, the primary focus of our work is to raise awareness regarding the importance of considering dynamic environments and tool variability in tool learning, rather than demonstrating whether adapter tuning or LoRA is superior to full parameter fine-tuning.
>
> In our study, we integrate environmental feedback and tool variability into the training episodes through mechanisms such as MCTS, self-reflection, and tool updates. This enables the model to recognize that tools may become outdated and to learn how to utilize new tools based on environmental feedback. Therefore, the choice between LoRA and full parameter fine-tuning is simply a different choice of training methodology. This is not the focal point of our work, and we do not find it necessary to substantiate the differences between these two approaches.
>
> Additionally, from a high-level design perspective, we would like to have more discussion on whether the foundational LLM should be modified. From our viewpoint, the ability of adaptation to dynamic environment is a critical foundational capability for a general AI model, and this ability should not be offloaded to external, manually designed frameworks or modules separate from the model itself.
>
> > W5
>
> We have provided clarifications regarding some of the issues you mentioned above, with the aim of enhancing your understanding of our work. Additionally, we seek to foster more substantive discussions that will strengthen our manuscript.

---

> > ### Author Response · Authors · 2024-11-15
> >
> > > Q1
> >
> > As mentioned in W1, we are unsure which aspects are perceived as inefficient in our work. We would greatly appreciate further feedback to facilitate constructive dialogue.
> >
> > > Q2
> >
> > Similar to our response in W4, our work focuses on highlighting the importance of dynamic environments and tool variability in tool learning, rather than establishing whether adapter tuning or LoRA is superior. We sincerely appreciate your comments and look forward to further discussions.
> >
> > > Q3
> >
> > As mentioned in `Line 316`, all methods in this study, including ours, utilize **greedy decoding** for inference to ensure a fair comparison. To further address your concerns, we provide the average response time compared to Static-SFT, acknowledging the limitations posed by proprietary models, such as queries per second (QPS) and network latency:
> >
> > | Method    | Time Per Question | Avg. Steps |
> > |-----------|-------------------|------------|
> > | Static-SFT| 7.8s              | 9.6        |
> > | ToolEVO   | 6.6s              | 7.4        |
> >
> > It is worth noting that ToolEVO demonstrates exceptional inference speed. Static-SFT, on the other hand, generates more incorrect calls due to its failure to adapt to tool variability, resulting in a greater number of inference steps.
> >
> > We believe our explanations have clarified your concerns and offer a clearer understanding of our work. We are eager to engage in further discussions to enhance and refine our work.
> >
> > Sincerely,
> >
> > Authors of 2051

---

> ### Author Response · Authors · 2024-11-20
>
> Dear Reviewer hYZV,
>
> We hope this message finds you well. We want to express our sincere gratitude for the thoughtful feedback you provided on our manuscript. We have carefully addressed each of your concerns in our rebuttal response and made substantial efforts to clarify the points you raised.
>
> Given that we are approaching the end of the discussion period, we would greatly value your thoughts on our responses. If you feel that our explanations have sufficiently addressed your concerns, we would be grateful if you could consider updating your evaluation. Of course, if you have any remaining questions or need further clarification, please don't hesitate to ask.
>
> Your expertise and insights are invaluable to us in improving our work.
> Thank you for your time and consideration.
>
> Best regards,
>
> Authors of 2051

---

> > ### Comment · Area_Chair_XmvY · 2024-11-24
> > **Please respond to the rebuttal ASAP**
> >
> > Dear reviewer,
> > The process only works if we engage in discussion. Can you please respond to the rebuttal provided by the authors ASAP?

---

### Meta-Review · Area_Chair_XmvY · 2024-12-20

**Metareview:**

This paper aims to develop an adaptive frameworks for LLMs to deal with tool variability. The key claim is that in the face of tool variability and change, the LLM can update it's usage pattern using a combination of self reflection and tool updating. The self reflection module uses a modified version of MCTS, while the tool update functionality is implemented via the LLM calling a system tool. The authors introduce a dynamic tool use environment and show the benefits of their methodology.

Strengths:
Interesting and timely topic
Reasonably novel and interesting methodology
Thorough experimental validation and strong ablation studies and analysis.

Weaknesses
The paper is written in a very confusing way, using pretty verbose and non-standard terminology. Some of this has been resolved through discussion, but the problem remains
The justification of the problem setting needs to be made stronger
The fact that MCTS Is not used at inference time needs to be made much more clear in the paper
Some of the baseline comparisons (eg with Claude) need to be made more clear as references rather than baselines.

**Additional Comments On Reviewer Discussion:**

The reviewers engaged in discussion mostly with the author, besides hYZV. The reviewers brought up a number of confusions about inference time, baselines, simple heuristic methods that could be used as comparisons. Two of the reviewers upped their score to a 6 due to changes implemented to address these.

Reviewer hYZV brought down their score but neither engaged in discussion nor provided concrete, substantive points. Their assessment was thus weighted down.

Reviewer zgUw kept their score at a 3, but on reading the discussion, I felt that the authors answered most of their questions well and with reasonable justification.

Given this discussion an the fact that the paper issues are largely clarity related (and already improving), I will suggest an accept (weak).

---

### Decision · Program_Chairs · 2025-01-22

Accept (Poster)